# Conduction velocity along a key white matter tract is associated with autobiographical memory recall ability

Ian A Clark[1], Siawoosh Mohammadi[2], Martina F Callaghan[1], Eleanor A Maguire[1]*

[1]Wellcome Centre for Human Neuroimaging, Department of Imaging Neuroscience, UCL Queen Square Institute of Neurology, University College London, London, United Kingdom; [2]Institute of Systems Neuroscience, University Medical Centre Hamburg-Eppendorf, Hamburg, Germany

**Abstract** Conduction velocity is the speed at which electrical signals travel along axons and is a crucial determinant of neural communication. Inferences about conduction velocity can now be made in vivo in humans using a measure called the magnetic resonance (MR) g-ratio. This is the ratio of the inner axon diameter relative to that of the axon plus the myelin sheath that encases it. Here, in the first application to cognition, we found that variations in MR g-ratio, and by inference conduction velocity, of the parahippocampal cingulum bundle were associated with autobiographical memory recall ability in 217 healthy adults. This tract connects the hippocampus with a range of other brain areas. We further observed that the association seemed to be with inner axon diameter rather than myelin content. The extent to which neurites were coherently organised within the parahippocampal cingulum bundle was also linked with autobiographical memory recall ability. Moreover, these findings were specific to autobiographical memory recall and were not apparent for laboratory-based memory tests. Our results offer a new perspective on individual differences in autobiographical memory recall ability, highlighting the possible influence of specific white matter microstructure features on conduction velocity when recalling detailed memories of real-life past experiences.

*For correspondence:
e.maguire@ucl.ac.uk

**Competing interest:** The authors declare that no competing interests exist.

## Editor's evaluation

In this paper, the authors show that autobiographical memory recall is related to a specific biophysical property of the parahippocampal cingulum bundle, the MR g-ratio. The paper presents compelling data supporting a fundamental new insight about the relationship between autobiographical memory and its underlying neural anatomy. The paper will be of strong interest to memory researchers as well as neuroscientists studying associations between brain structure and cognitive processes more generally.

## Introduction

Communication between neurons in the brain is critical for cognition, and depends upon action potentials being conveyed along axons within white matter tracts. The speed at which these electrical signals travel along axons is known as the conduction velocity. It has been suggested that faster axonal conduction velocity promotes better cognition. For example, an increase in axonal conduction velocity is hypothesised to underpin the greater cognitive processing ability of vertebrates, in particular primates and humans (*Brancucci, 2012*; *Miller, 1994*), compared to invertebrates (*Arancibia-Cárcamo et al., 2017*; *Bullock et al., 1984*; *Nave, 2010*). In rats (*Aston-Jones et al., 1985*) and cats

(*Xi et al., 1999*), faster axonal conduction velocity has been observed in younger compared to older animals. In a similar vein, axonal degradation that can lead to reduced conduction velocity has been identified in older compared to younger monkeys (*Peters et al., 2000*; *Peters and Sethares, 2002*).

Echoing these findings from non-humans, estimates of conduction velocity in humans from the latency of visual evoked potentials recorded over primary visual cortex (*Reed and Jensen, 1992*) and between the thalamus and parietal cortex (*Reed and Jensen, 1993*) have been positively correlated with nonverbal intelligence quotients. In addition, faster axonal conduction velocities are thought to better explain increases in intelligence compared to absolute and relative brain volumes (*Dicke and Roth, 2016*).

The conduction velocity of an axon is dependent upon the axon diameter, the presence and thickness of a myelin sheath, the distance between the nodes of Ranvier (periodic gaps in the myelin that facilitate action potential propagation), inter-nodal spacing, and electrical properties of the axonal and myelin membranes (*Arancibia-Cárcamo et al., 2017*; *Drakesmith et al., 2019*; *Gasser and Grundfest, 1939*; *Hursh, 1939*; *Huxley and Stampfli, 1949*; *Rushton, 1951*). A number of these features are not yet measurable in humans in vivo. However, seminal electrophysiological work has derived a relationship between axon morphology and conduction velocity using only axon diameter and myelin sheath thickness (*Rushton, 1951*). These two metrics are particularly key because a larger axon diameter results in less resistance to the action potential ion flow, resulting in faster conduction velocity. The presence of a myelin sheath around an axon is beneficial in two ways. First, the myelin sheath acts like an electrical insulating layer, reducing ion loss and preserving the action potential. Second, the presence of unmyelinated gaps in the myelin sheath (the nodes of Ranvier) enables a process called saltatory propagation to take place. As the majority of the axon is wrapped in myelin, the nodes of Ranvier are the only locations where action potentials can occur. This increases the strength of electrical signals because all the ions gather at these nodes instead of being dispersed along the length of the axon. Stronger action potentials are therefore sent along the myelinated portion of the axon at higher speeds, with this signal being boosted on arrival at the next node of Ranvier by another action potential, which helps to maintain a fast conduction velocity.

The conduction velocity of an axon is, therefore, not determined only by the axon diameter, but also by the relationship between the axon diameter and the thickness of the surrounding myelin sheath, a measure known as the g-ratio (*Chomiak and Hu, 2009*; *Rushton, 1951*; *Schmidt and Knösche, 2019*). Specifically, *Rushton, 1951* derived an equation: conduction velocity $\propto d\sqrt{-\ln(g)}$ where d is the inner axon diameter, g is the g-ratio=d/D, and D is the outer fibre (axon plus myelin sheath) diameter. In other words, the g-ratio is computed as the ratio of the inner axon diameter relative to that of the axon plus the myelin sheath that encases it (*Figure 1*).

Until recently, g-ratio measurements were restricted to invasive studies in non-human animals. However, by combining diffusion magnetic resonance imaging (MRI) with quantitative structural MRI scans optimised to assess myelination (e.g. magnetisation transfer saturation; *Weiskopf et al., 2013*), it is now possible to estimate the g-ratio in vivo in humans across the whole brain (*Drakesmith et al., 2019*; *Mohammadi et al., 2015*; *Mohammadi and Callaghan, 2021*; *Stikov et al., 2015*). This is

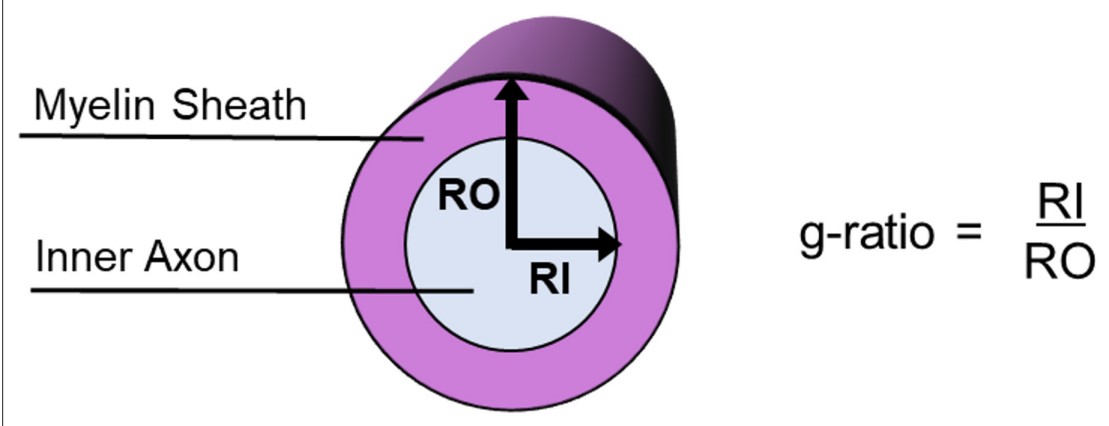

**Figure 1.** Schematic of a myelinated axon showing how the g-ratio is calculated.

achieved by measuring an aggregate g-ratio, which is an area-weighted ensemble average across a voxel of an underlying distribution of microscopic g-ratios of axons (*Stikov et al., 2015*; *West et al., 2016*). Whole brain MR g-ratio maps enable the investigation of the MR g-ratio of white matter fibre pathways at the group level. These MR g-ratio estimates have been optimised (*Ellerbrock and Mohammadi, 2018*; *Jung et al., 2018*; *West et al., 2018*) and used to investigate white matter development (*Cercignani et al., 2017*), changes in the g-ratio during aging (*Berman et al., 2018*), and as a potential neuroimaging marker in patients with multiple sclerosis (*Yu et al., 2019*). Of note, the MR g-ratio has been found to associate well with estimates of axonal conduction velocity (*Berman et al., 2019*; *Drakesmith et al., 2019*). Consequently, the MR g-ratio provides a non-invasive MRI method that can associate in vivo structural neuroimaging of humans with axonal conduction velocity.

Given that the MR g-ratio encompasses information about the inner axon diameter and myelin thickness, the identification of statistically significant relationships with the MR g-ratio can also guide as to which of these two features might be influencing variations in conduction velocity (*Caeyenberghs et al., 2016*; *Kaller et al., 2017*; *Lakhani et al., 2016*; *Waxman, 1980*; *Xin and Chan, 2020*). This is particularly useful in relation to the inner axon diameter. While measures of myelination can be obtained using conventional MRI scanners (e.g. via magnetisation transfer saturation maps), estimates of inner axon diameter cannot, and instead require ultra-strong gradient systems – 300 mT/m compared to the 10s of mT/m of conventional MRI scanners (*Jones et al., 2018*; *Veraart et al., 2020*). However, only four ultra-strong gradient Connectom MRI scanners exist in the world. Consequently, measuring the MR g-ratio with conventional MRI scanners can be useful in guiding inferences about inner axon diameter in the context of conduction velocity.

Adjudicating between the possible influence of inner axon diameter or myelination relies on knowledge, or an assumption, about whether the associated change in conduction velocity is faster or slower. This is because significant associations with the MR g-ratio only indicate the existence of a relationship with conduction velocity but not the direction. As noted previously, faster conduction velocity is often held to promote better cognition (e.g. *Brancucci, 2012*; *Dicke and Roth, 2016*; *Miller, 1994*; *Reed and Jensen, 1992*). Therefore, in *Figure 2A*, inner axon diameter, myelin thickness and (MR) g-ratio are plotted together to illustrate how different changes in the MR g-ratio are related to the underlying microstructural properties, given a faster conduction velocity. Myelin thickness is represented by the gradient in background colour and contours, with thinnest myelin at the bottom right, and thickest on the top left. The direction of the arrows describes the change in g-ratio for the microstructural variations presented. There are three main scenarios. (1) A decrease in MR g-ratio values. This would suggest that faster conduction velocity is due to greater thickness of the myelin sheath, with the inner axon diameter remaining constant (*Figure 2A* blue arrow, and *Figure 2B*). (2) A decrease in MR g-ratio values, but to a lesser extent than that observed in the first scenario. This would suggest that faster conduction velocity is due primarily to greater myelin sheath thickness, but one that is also accompanied by a larger inner axon diameter (*Figure 2A* red arrow, and *Figure 2C*). (3) An increase in MR g-ratio values. This would suggest that faster conduction velocity is predominantly due to a larger inner axon diameter, with only small differences, if any, in myelin thickness being present (*Figure 2A* orange arrow, and *Figure 2D*).

We note, for completeness, that a fourth scenario also exists where constant MR g-ratio values could be associated with faster conduction velocity. This would occur when both the inner axon diameter and myelin thickness change proportionally to each other (*Figure 2A* black arrow, and *Figure 2E*). However, this scenario could also mean that there is no variation in conduction velocity. One way to increase interpretability in this situation is by examining scans optimised to assess myelination (such as magnetisation transfer saturation values). Observing no relationship with the MR g-ratio, but a relationship with magnetisation transfer saturation values, would suggest a proportional change in the underlying myelin and, consequently, variation in conduction velocity. By contrast, no relationship with either the MR g-ratio or magnetisation transfer saturation values would suggest that there was no change in the underlying microstructure and therefore no variation in conduction velocity.

The MR g-ratio has, therefore, the potential to provide a number of novel insights into human cognition. One area where the MR g-ratio may be particularly helpful is in probing individual differences. Our particular interest is in the ability to recall past experiences from real life, known as autobiographical memories. The detailed recall of autobiographical memories is a critical cognitive function that serves to sustain our sense of self, enable independent living, and prolong survival (*Tulving,*

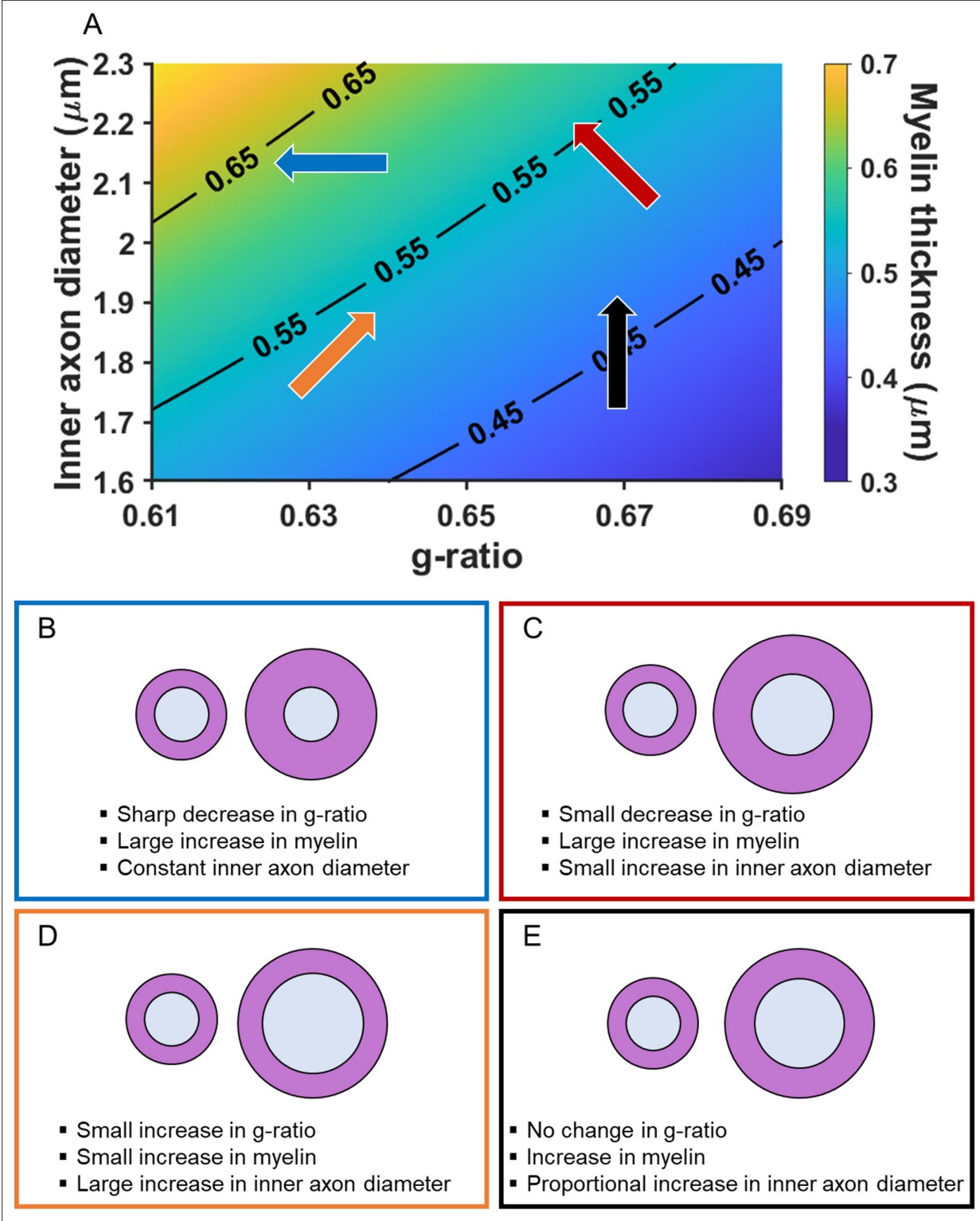

**Figure 2.** Illustration of how the MR g-ratio relates to specific microstructural properties given a faster conduction velocity. (**A**) Graphical representation of the relationships between myelin thickness, inner axon diameter and MR g-ratio, assuming a faster conduction velocity (see Appendix 1 for details of the simulation). Myelin thickness is represented by the gradient in background colour and contours on the graph, with the thinnest myelin at the bottom right and thickest at the top left. The direction of the arrows describes the change in g-ratio for each microstructural variation presented in B-E. The positioning and colours of the arrows correspond to the text box outline colours in B-E. (**B–E**) Illustrations of how changes in the MR g-ratio relate to the underlying axonal microstructure, given a faster conduction velocity.

*2002*). While some healthy individuals can recollect decades-old autobiographical memories with great richness and clarity, others struggle to recall what they did last weekend (*LePort et al., 2012*; *Palombo et al., 2015*). In the context of the healthy population, we currently lack a clear biological explanation for the basis of these individual differences (*Palombo et al., 2018*).

There is no doubt that the hippocampus is central to the processing of autobiographical memories, and hippocampal damage is linked with autobiographical memory impairments (*McCormick et al., 2018*; *Scoville and Milner, 1957*; *Winocur and Moscovitch, 2011*). However, no consistent relationship between autobiographical memory recall ability and hippocampal grey matter volume or microstructure has been identified in healthy individuals (*Clark et al., 2020*; *Clark et al., 2021a*; *LePort et al., 2012*; *Maguire et al., 2003*; *Van Petten, 2004*). The hippocampus does not act alone, and functional neuroimaging studies have revealed that a distributed set of brain areas supports autobiographical memory recall along with the hippocampus, including the parahippocampal, retrosplenial,

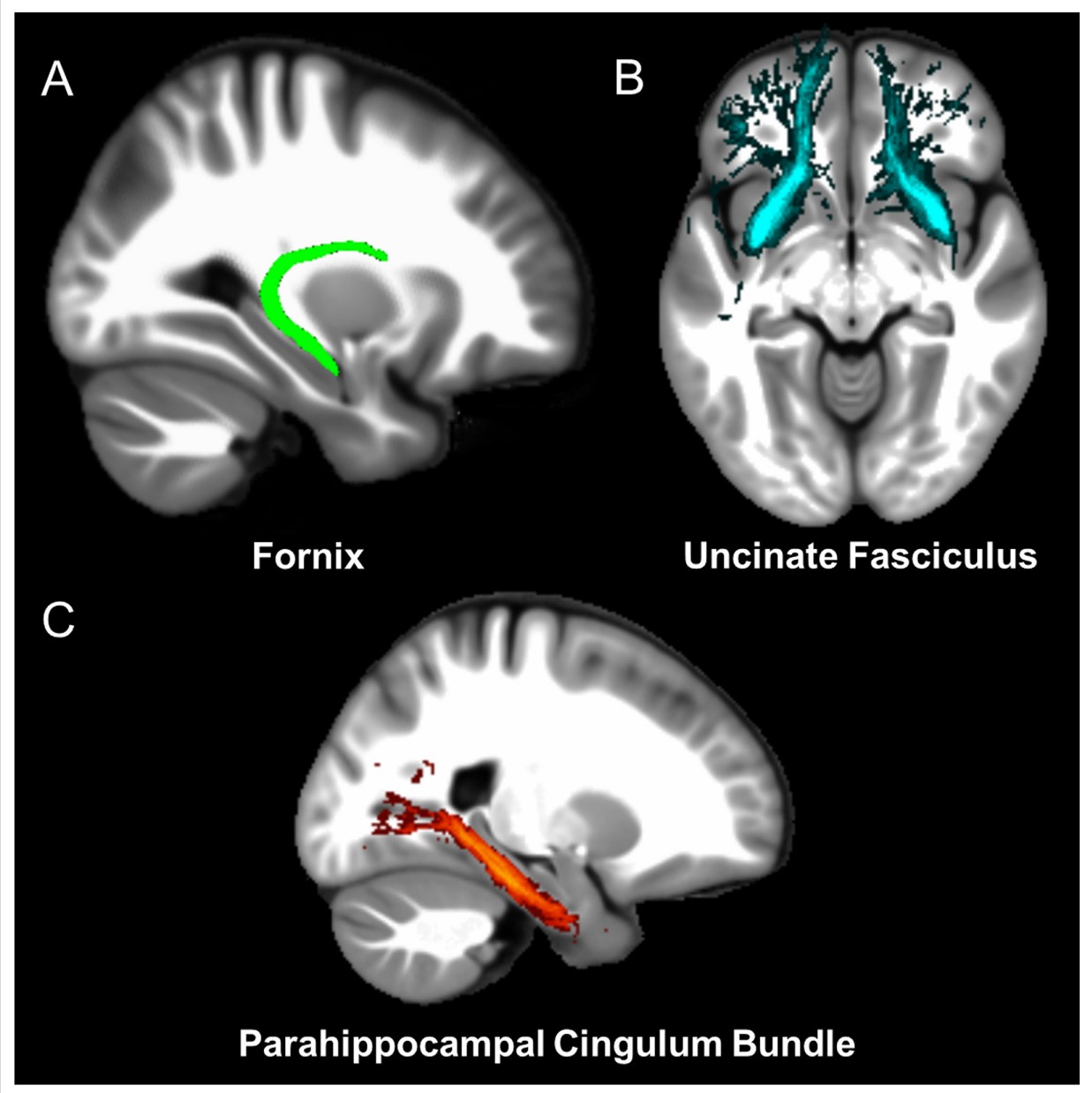

**Figure 3.** The three white matter tracts of interest, given their relationship with the hippocampal region. The fornix was defined using the ICBM-DTI-81 white-matter labels atlas (*Mori et al., 2008*). The uncinate fasciculus and parahippocampal cingulum bundle were defined using the Johns Hopkins probabilistic white matter tractography atlas (*Hua et al., 2008*), with the minimum probability threshold set to 25%.

parietal and medial prefrontal cortices (*Andrews-Hanna et al., 2014*; *Maguire, 2001*; *Spreng et al., 2009*; *Svoboda et al., 2006*). Changes in MR g-ratio and, by inference, conduction velocity, might affect communication between these brain regions and so influence individual differences in autobiographical memory recall within the healthy population.

Three white matter pathways in particular enable communication with the hippocampal region – the fornix, the uncinate fasciculus and the parahippocampal cingulum bundle. The fornix (*Figure 3A*) is a major pathway in and out of the hippocampus and connects it to the orbital and medial prefrontal cortices, the basal forebrain, the anterior thalamus, the hypothalamus and the mammillary bodies (*Aggleton et al., 2015*; *Croxson et al., 2005*). The uncinate fasciculus (*Figure 3B*) originates in the uncus, entorhinal and parahippocampal cortices and passes over the lateral nucleus of the amygdala, arcs around the Sylvian fissure, terminating in various locations throughout the prefrontal cortex (*Croxson et al., 2005*; *Von Der Heide et al., 2013*). The parahippocampal cingulum bundle (*Figure 3C*) links the hippocampus with the entorhinal, parahippocampal, retrosplenial and parietal cortices, as well as providing another route between the hippocampus and anterior thalamus (*Bubb et al., 2018*; *Jones et al., 2013b*). It also links to the prefrontal cortex via its connections with other parts of the cingulum bundle.

In the current study, we calculated the MR g-ratio within these three pathways to ascertain whether this was related to autobiographical memory recall ability. A significant relationship with the MR g-ratio would suggest that variation in the conduction velocity of a pathway is associated with autobiographical memory recall ability. The relationship of MR g-ratio to the underlying microstructure as outlined in *Figure 2*, would further guide us as to whether any significant effects were more likely to be associated with the extent of myelination or the size of the inner axon diameter of the fibres in these three white matter tracts. These analyses were augmented by examining whether magnetisation transfer saturation values of the pathways (assessing myelination) were associated with autobiographical memory recall ability.

In addition, we also used the neurite orientation dispersion and density imaging (NODDI; *Zhang et al., 2012*) biophysical model to derive two complementary biological measures that could provide further insights into the arrangement of neurites in a voxel. A neurite is any projection from a neuron's cell body, such as an axon or a dendrite. The neurite orientation dispersion index is an estimate of the organisation of the neurites in a voxel, where a small orientation dispersion index value indicates a low dispersion of neurites, in other words, that the neurites are coherently organised. The second property, neurite density, is a measure of the density of the neurites in a voxel.

For completeness, the commonly reported physical parameters from standard diffusion tensor imaging (DTI; e.g. fractional anisotropy and mean diffusivity; *Basser, 1995*) that are often derived from diffusion data were also computed (*Oeschger et al., 2021*). However, these metrics lack biological specificity (*Jensen and Helpern, 2010*; *Jones et al., 2013a*) and, consequently, could not speak to our research questions. For example, fractional anisotropy is a very general measure. Variations in fractional anisotropy values can occur for numerous reasons, including, but not limited to, changes in the extent of myelination, axon coherence, axon density, and the level of astrocytes. Consequently, standard DTI parameter results are only briefly summarised in the main text, with full details provided in Appendix 1 and *Supplementary file 1*, *Supplementary file 2*.

To ensure an appropriate sample size and a wide range of autobiographical memory recall ability, we examined a large group of healthy young adults from the general population (n=217; 109 females, 108 males; mean age of 29.0 years, SD = 5.60; age was restricted to between 20 and 41 years to limit the possible effects of aging).

All participants underwent the widely-used Autobiographical Interview (*Levine et al., 2002*). This task, which is the gold standard in the field, provides a detailed metric characterising a person's ability to recall real-life past experiences, as well as providing a useful control measure. We also examined performance on another eight standard laboratory-based memory tests. Their inclusion allowed us to ascertain whether any relationships with conduction velocity were specific to recollecting detailed autobiographical memories from real life, or were also applicable to the recall of more constrained laboratory-based stimuli.

Diffusion and magnetisation transfer saturation MRI scans were obtained for each person to enable calculation of the MR g-ratio and the other measures. Our analyses were performed using weighted means from each of the three white matter tracts of interest rather than voxel-wise across the whole brain, reducing the potential for false positives (*Marek et al., 2022*).

Focusing on the fornix, uncinate fasciculus and parahippocampal cingulum bundle, we predicted that variations in the MR g-ratio from some or all of these tracts would be associated with autobiographical memory recall ability. Such a finding would, for the first time, suggest a link between variations in white matter tract conduction velocity and individual differences in autobiographical memory recall.

Under the assumption that a significant relationship between the MR g-ratio and autobiographical memory recall represents an association with faster conduction velocity (e.g. *Brancucci, 2012*; *Dicke and Roth, 2016*; *Miller, 1994*; *Reed and Jensen, 1992*), and aided by the analysis of the magnetisation transfer saturation values, we further sought to evaluate the scenarios presented in *Figure 2*. Specifically, we asked whether autobiographical memory recall ability was more likely to be associated with inner axon diameter or myelin thickness in the context of g-ratio and conduction velocity. A negative relationship between the MR g-ratio and autobiographical memory recall ability (*Figure 2A* blue and red arrows, and *Figure 2B and C*), along with a positive relationship between magnetisation transfer saturation and autobiographical memory recall ability, would suggest that myelin thickness was more relevant. By contrast, a positive relationship between the MR g-ratio and autobiographical memory recall ability (*Figure 2A* orange arrow, and *Figure 2D*) would highlight the potential relevance of inner axon diameters. Observing no associations between the MR g-ratio or magnetisation transfer saturation values and autobiographical memory recall, would suggest no relationships with the underlying microstructure or conduction velocity, and would speak against the possibility of a proportional change in microstructure (*Figure 2A* black arrow, and *Figure 2E*).

Finally, given that a previous functional MRI meta-analysis identified different neural substrates associated with autobiographical memory recall and the recall of laboratory-based stimuli (*McDermott et al., 2009*; see also *Maguire, 2001*; *Maguire, 2012*; *Miller et al., 2022*; *Mobbs et al., 2021*; *Nastase et al., 2020*; *Spiers and Maguire, 2007*), we expected that associations with the MR g-ratio might be specific to autobiographical memory recall. We reasoned that detailed, multimodal, autobiographical memories may rely on inter-regional connectivity to a greater degree than simpler, more constrained laboratory-based memory tests.

## Results

### Autobiographical memory recall scores

We employed the widely-used Autobiographical Interview (*Levine et al., 2002*) to score autobiographical memory recall (see Materials and methods for full details). The main measure of autobiographical memory recall ability was the mean number of 'internal' details from the freely recalled autobiographical memories. Internal details are those that describe the specific past event in question, and are considered to reflect episodic information. Across the participants, the mean number of internal details provided per memory was 23.95 (SD = 7.25; range = 4.60–44.60).

As a control measure, the mean number of "external" details was also calculated from the autobiographical memory descriptions. External details pertain to semantic information about the past event, and other non-event information. Across the participants, the mean number of external details provided per memory was 5.35 (SD = 3.20; range = 0.8–17.40).

### Laboratory-based memory test performance

While our main interest was in autobiographical memory recall, eight commonly used laboratory-based memory tasks were also administered. Their inclusion allowed us to establish whether any associations identified with the main microstructure measure, MR g-ratio, were specific or not to the recollection of real-life autobiographical memories.

The ability to recall a short narrative was examined using the immediate and delayed recall tests of the Logical Memory subtest of the Wechsler Memory Scale IV (*Wechsler, 2009*). Across participants, the mean immediate recall scaled score was 12.95 (SD = 2.09, range = 6–18) and the mean delayed recall scaled score was 12.58 (SD = 2.62, range = 6–19). Verbal list recall ability was assessed using the immediate and delayed recall of the Rey Auditory Verbal Learning Test (see *Strauss et al., 2006*). The mean immediate recall (aggregate) score was 58.82 (SD = 7.42, range = 33–73) and the mean delayed recall score was 12.92 (SD = 2.17, range = 6–15). Visuospatial recall ability was examined using the delayed recall of the Rey–Osterrieth Complex Figure (*Rey, 1941*), with a mean delayed recall score

of 22.28 (SD = 5.71, range = 8.5–35). Recognition memory ability was tested using the Warrington Recognition Memory Tests for Words and Faces (*Warrington, 1984*). The mean recognition memory scaled score for words was 12.75 (SD = 2.06, range = 3–15) and for faces was 11.00 (SD = 3.33, range = 3–18). Finally, participants also completed the 'Dead or Alive' task which probes general knowledge about whether famous people have died or are still alive, providing a measure of semantic memory (*Kapur et al., 1989*). The mean accuracy performance on this test was 81.32% (SD = 8.44, range = 57.14%–97.26%).

**Table 1.** Means and standard deviations for the microstructure measures from the parahippocampal cingulum bundle.

| Microstructure measure | Mean | Standard deviation |
|---|---|---|
| MR g-ratio | 0.647 | 0.043 |
| Magnetisation transfer saturation | 0.959 | 0.007 |
| Neurite dispersion (ODI) | 0.189 | 0.038 |
| Neurite density | 0.480 | 0.051 |

Note. ODI = Orientation Dispersion Index.

## No relationships between memory measures and fornix or uncinate fasciculus microstructure

We first investigated the fornix and uncinate fasciculus. None of the biophysical measures from either tract were significantly associated with autobiographical memory recall ability. This was the case when using a corrected $p < 0.017$ (see Materials and methods) or an uncorrected $p < 0.05$ threshold. Full details of these results are provided in Appendix 1 (*Appendix 1—figures 1 and 2*; *Appendix 1—tables 1–4*), with the source data available in *Supplementary file 1*.

Standard DTI parameters (e.g. fractional anisotropy and mean diffusivity) were also extracted from the fornix and uncinate fasciculus. None of the standard DTI parameters, from either tract, were significantly associated with autobiographical memory recall ability, even when using an uncorrected $p < 0.05$ threshold (see *Appendix 1—tables 1–4* for full details, source data are available in *Supplementary file 1*).

There were no significant associations between fornix or uncinate fasciculus MR g-ratios or magnetisation transfer saturation values and any of the laboratory-based memory tests (see *Appendix 1— table 5*, *Appendix 1—table 6* for full details, source data are available in *Supplementary file 1*).

Of note, and for completeness, we also performed exploratory analyses in six additional white matter tracts: the anterior thalamic radiation, the dorsal cingulum bundle, the forceps minor, the

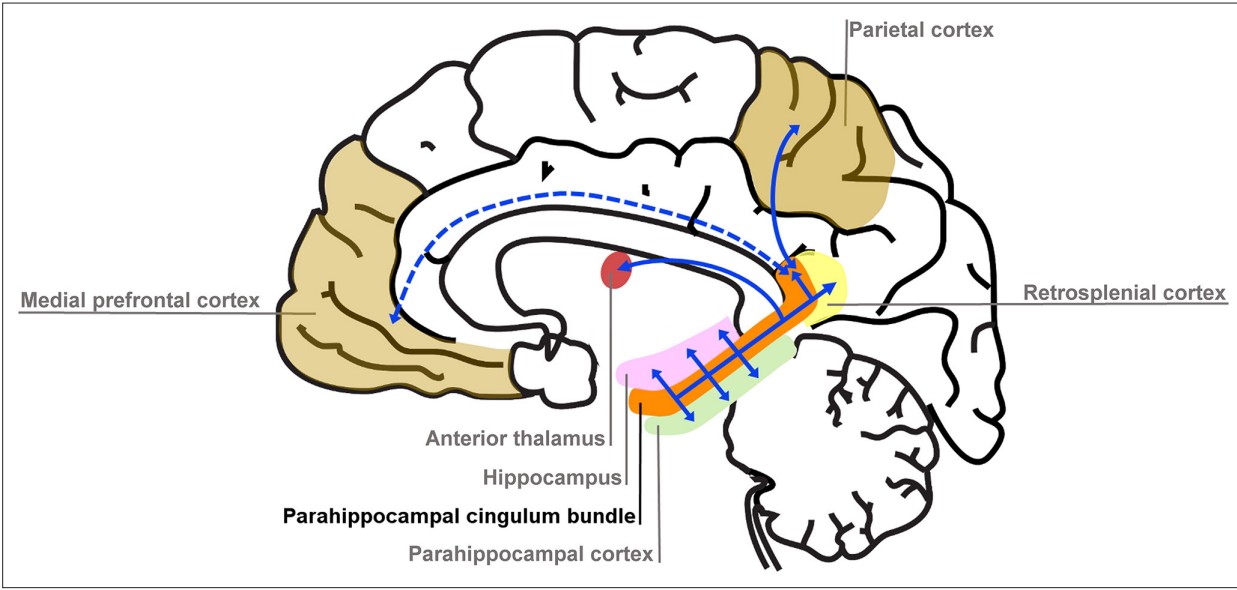

**Figure 4.** Simplified schematic of the location and main connections of the parahippocampal cingulum bundle. The blue lines indicate direct connections, and the dashed blue line an indirect connection.

inferior longitudinal fasciculus, the inferior occipitofrontal fasciculus and the superior longitudinal fasciculus. However, as with the fornix and the uncinate fasciculus, none of the metrics from any of these tracts were associated with autobiographical memory recall ability, even when using an uncorrected p<0.05 threshold (see *Appendix 1—figures 3–8* and *Appendix 1—tables 7–18* for full details).

### The parahippocampal cingulum bundle

We found that variations in autobiographical memory recall ability were uniquely related to the microstructure of the parahippocampal cingulum bundle. This tract connects the hippocampus with the entorhinal, parahippocampal, retrosplenial and parietal cortices, and the anterior thalamus (*Figure 4*). Moreover, via other subdivisions of the cingulum bundle, it is indirectly connected with prefrontal regions including the medial prefrontal cortex. The parahippocampal cingulum bundle is, therefore, well positioned for information transfer between the key regions involved in autobiographical memory recall (*Andrews-Hanna et al., 2014*; *Maguire, 2001*; *Spreng et al., 2009*; *Svoboda et al., 2006*).

As with the other tracts (see Materials and methods), the parahippocampal cingulum bundle region of interest (ROI) was defined bilaterally using the Johns Hopkins probabilistic white matter tractography atlas (*Hua et al., 2008*). To reduce partial volume effects, we used a conservative minimum probability of 25%, and the tract ROI was refined for each participant to ensure the mask was limited to each person's white matter. The mean number of voxels in the parahippocampal cingulum bundle ROI was 129.11 (SD = 25.68), and the variance in number of voxels across individuals was accounted for in our analyses. As before, a corrected statistical threshold of p<0.017 was applied (see Materials and methods). *Table 1* shows the summary statistics for the microstructure measures, with the source data for the parahippocampal cingulum bundle available in *Supplementary file 2*.

### Variation in MR g-ratio of the parahippocampal cingulum bundle was associated with autobiographical memory recall ability

We first investigated whether the MR g-ratio of the parahippocampal cingulum bundle was associated with autobiographical memory recall ability, performing partial correlation analyses with age, gender, scanner and the number of voxels in the ROI included as covariates. A significant positive association was observed between the parahippocampal cingulum bundle MR g-ratio and the number of internal details (*Figure 5A*; $r(211) = 0.18$, p=0.008, 95% CI=0.05, 0.29). This relationship was specific to internal details, with no association evident for the external details control measure (*Figure 5B*; $r(211) = –0.09$, p=0.17, 95% CI=−0.21, 0.019). Direct comparison of the correlations confirmed there was a significantly larger correlation between the MR g-ratio and internal details than for external details (*Figure 5C*; mean r difference = 0.28 (95% CI=0.11, 0.45), $z$=3.21, p=0.0013). This suggests that variations in the conduction velocity of the parahippocampal cingulum bundle are associated with individual differences in autobiographical memory recall ability. The positive nature of the correlation highlights the potential relevance of inner axon diameter rather than myelin thickness in the context of g-ratio and conduction velocity (*Figure 2A* orange arrow, and *Figure 2D*).

To further aid the interpretation of the relationship with MR g-ratio in terms of the underlying axonal microstructure, we also investigated whether the magnetisation transfer saturation values (assessing myelination) of the parahippocampal cingulum bundle were associated with autobiographical memory recall ability. We performed partial correlation analyses with the same covariates as before. Magnetisation transfer saturation values were not significantly related to either internal ($r(211) = 0.00$, p=1.0, 95% CI=−0.12, 0.12), or external ($r(211) = 0.14$, p=0.048, 95% CI=−0.06, 0.30) details.

Overall, therefore, under the assumption that the significant relationship between the MR g-ratio and autobiographical memory recall ability is associated with faster conduction velocity, Rushton's model of conduction velocity would indicate that better autobiographical memory recall is more likely to be related to parahippocampal cingulum bundle axons having larger inner axon diameters rather than thicker myelin sheaths (see *Figure 2A* orange arrow, and *Figure 2D*).

### Variation in neurite dispersion within the parahippocampal cingulum bundle was associated with autobiographical memory recall

In addition to the MR g-ratio, we also examined the relationship between autobiographical memory recall ability and two complementary biophysical measures, the neurite orientation dispersion index and neurite density maps estimated using the NODDI biophysical model (*Zhang et al., 2012*). Partial

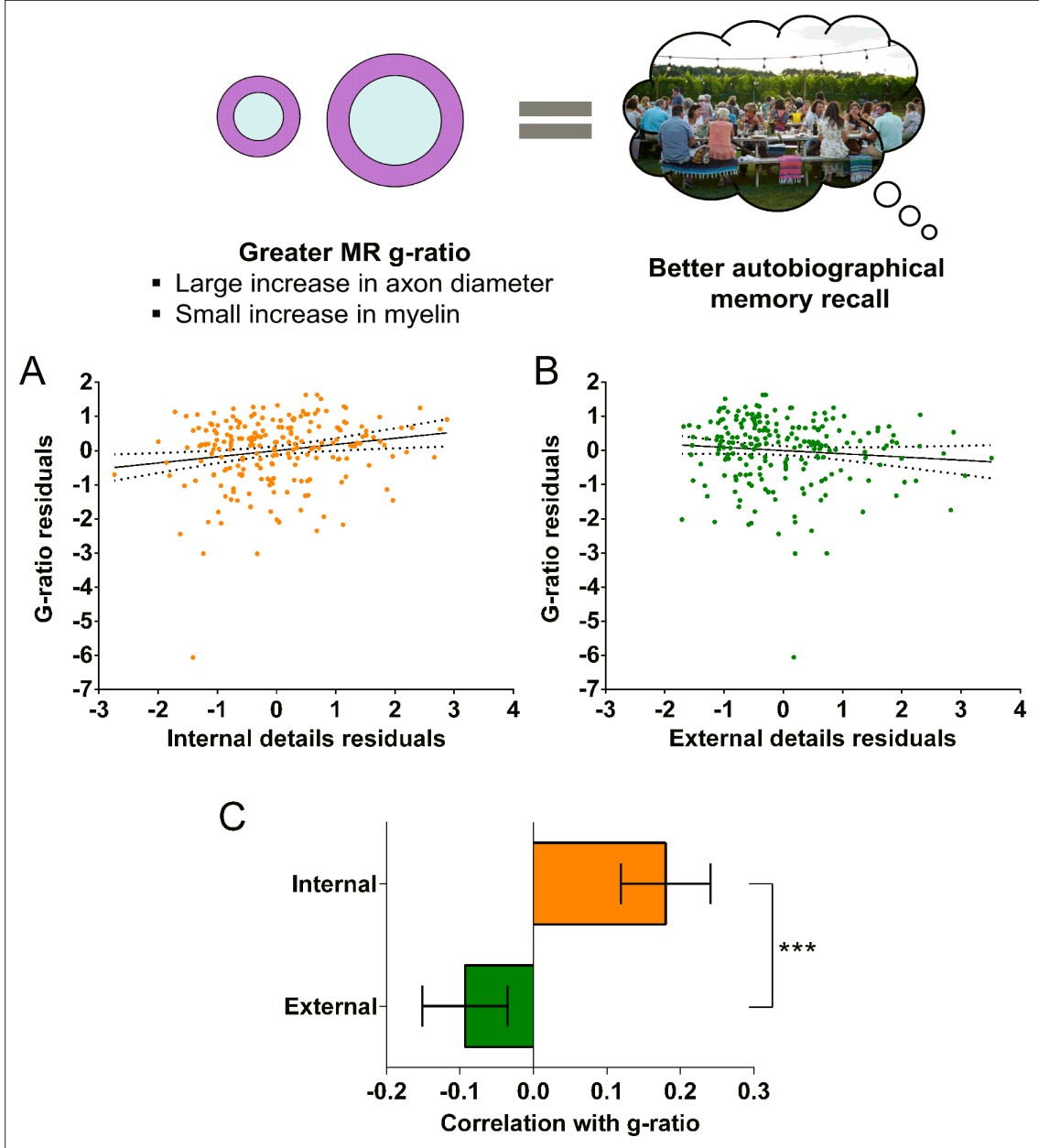

**Figure 5.** MR g-ratio and the parahippocampal cingulum bundle. The relationship between parahippocampal cingulum bundle MR g-ratio and autobiographical memory recall ability (internal details), and the control measure (external details) are shown. (**A**) There was a significant positive correlation between the MR g-ratio and internal details (dashed lines indicate the confidence intervals). (**B**) There was no significant relationship between the MR g-ratio and external details. (**C**) Bar chart showing the partial correlation coefficients (with standard errors) between the MR g-ratio and internal and external details. There was a significant difference between the correlations when they were directly compared; ***p<0.001. Data points for this figure are provided in *Figure 5—source data 1*, n = 217 for all analyses.

The online version of this article includes the following source data for figure 5:

**Source data 1.** Source data for the data points in *Figure 5*.

correlations revealed a significant negative correlation between the neurite orientation dispersion index (a small orientation dispersion index value indicates low dispersion) and internal details (*Figure 6A*; $r(211) = -0.19$, p=0.005, 95% CI=−0.32,−0.06). This was again specific to internal details with no significant relationship between the neurite orientation dispersion index and external details (*Figure 6B*; $r(211) = 0.07$, p=0.28, 95% CI=−0.05, 0.20). Direct comparison of the correlations revealed a significantly larger correlation between the neurite orientation dispersion index and internal details

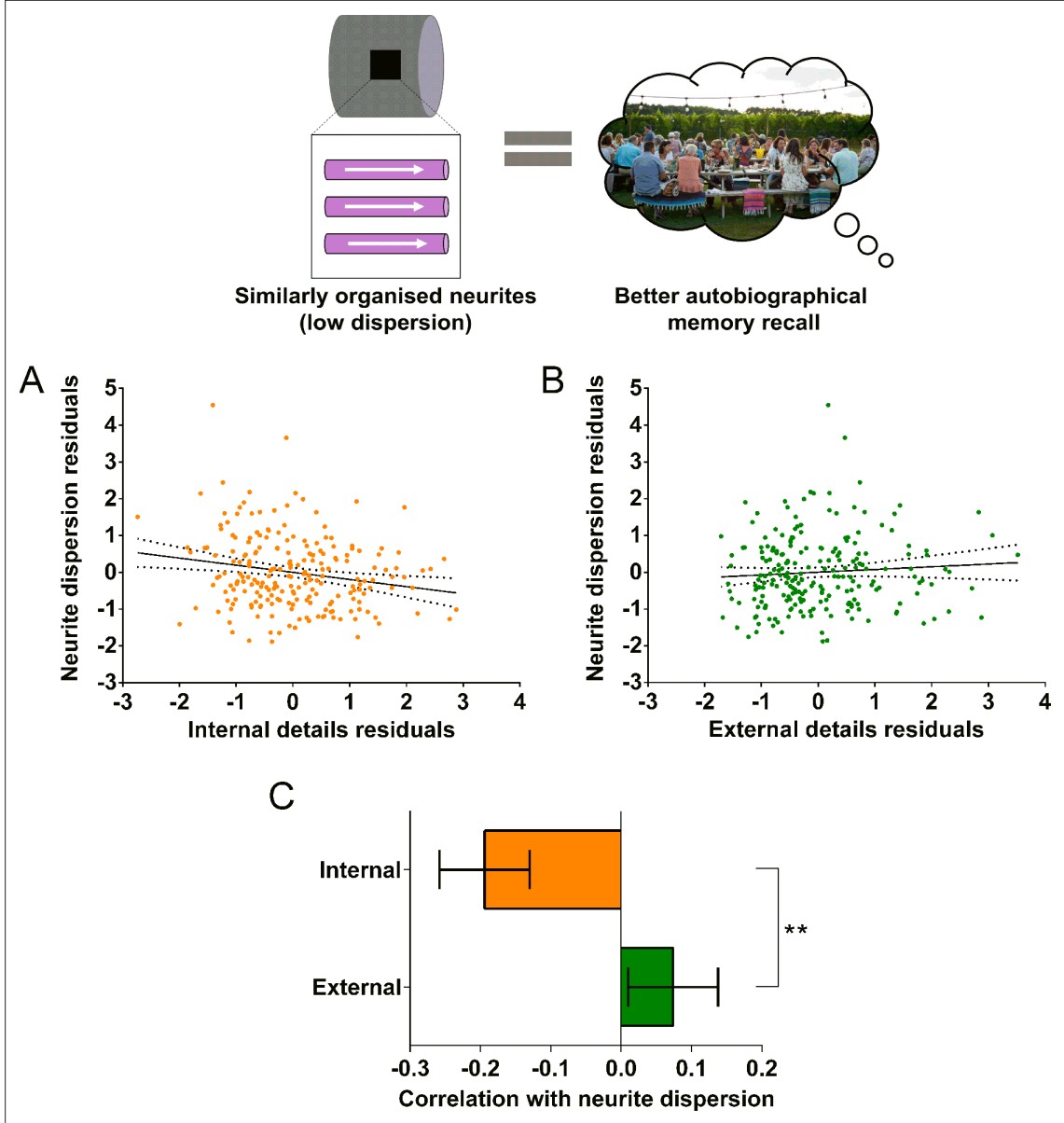

**Figure 6.** Neurite dispersion and the parahippocampal cingulum bundle. The relationship between parahippocampal cingulum bundle neurite dispersion (orientation dispersion index) and autobiographical memory recall ability (internal details), and the control measure (external details) are shown. (**A**) There was a significant negative correlation between neurite dispersion and internal details (dashed lines indicate the confidence intervals). (**B**) There was no significant relationship between neurite dispersion and external details. (**C**) Bar chart showing the partial correlation coefficients (with standard errors) between neurite dispersion and internal and external details. There was a significant difference between the correlations when they were directly compared; **p<0.01. Data points for this figure are provided in the *Figure 6—source data 1*, n = 217 for all analyses.

The online version of this article includes the following source data for figure 6:

**Source data 1.** Source data for the data points in *Figure 6*.

than for external details (*Figure 6C*; mean r difference = 0.27 (95% CI=0.44, 0.10), z=3.13, p=0.0017). Neurite density was not significantly related to either internal (r(211) = 0.04, p=0.60, 95% CI=−0.09, 0.16), or external (r(211) = 0.01, p=0.93, 95% CI=−0.12, 0.13) details. Therefore, in addition to a higher MR g-ratio, when neurites in the parahippocampal cingulum bundle were less dispersed and thus more coherently organised, this was associated with better autobiographical memory recall.

For completeness, we also examined a third measure from the NODDI biophysical model, the isotropic volume fraction. This models the space occupied by cerebrospinal fluid in a voxel, and therefore no relationship with autobiographical memory recall ability was expected, and none was found.

Partial correlations showed that the isotropic volume fraction of the parahippocampal cingulum bundle (mean = 0.04, SD = 0.03) was not significantly related to either internal ($r(211)$ = –0.03, $p=0.71$, 95% CI=−0.14, 0.10), or external ($r(211)$ = 0.02, $p=0.77$, 95% CI=−0.11, 0.15) details. This confirms that it is the combination of these measures, in the form of the MR g-ratio, that is meaningful, rather than any single measure alone.

## Standard DTI parameters of the parahippocampal cingulum bundle and autobiographical memory recall ability

Partial correlations using the same covariates as the previous analyses were also performed using the standard DTI parameters. As these metrics lack biological specificity (*Jensen and Helpern, 2010*; *Jones et al., 2013a*), and could not speak to our research questions, the results are summarised here, with full details in *Appendix 1—table 19*, *Appendix 1—table 20* and *Appendix 1—figure 9*, *Appendix 1—figure 10* and the source data are available in *Supplementary file 2*.

Two significant relationships were identified between the standard DTI parameters and autobiographical memory recall ability. First, there was a positive correlation between fractional anisotropy and the number of internal details (*Appendix 1—figure 9A*; $r(211)$ = 0.20, $p=0.003$, 95% CI=0.07, 0.32). Second, a positive correlation was evident between diffusivities parallel and the number of internal details (*Appendix 1—figure 10A*; $r(211)$ = 0.19, $p=0.005$, 95% CI=0.06, 0.32). As with the microstructure metrics, these relationships were specific to internal details with no relationships identified with external details for either fractional anisotropy ($r(211)$ = –0.06, $p=0.39$, 95% CI=−0.19, 0.07) or diffusivities parallel ($r(211)$ = –0.048, $p=0.49$, 95% CI=−0.19, 0.10). No significant relationships were found between mean diffusivity, mean kurtosis or diffusivities perpendicular and either internal or external details.

## No relationship between microstructure measures of the parahippocampal cingulum bundle and scores on laboratory-based memory tests

Finally, we tested for associations between the key microstructure measures and performance on the eight laboratory-based memory tasks. No relationships with the MR g-ratio or magnetisation transfer saturation values were evident for any task, even when using an uncorrected $p < 0.05$ threshold (*Appendix 1—table 21*, source data are available in *Supplementary file 2*). This suggests that the parahippocampal cingulum bundle MR g-ratio, and by inference conduction velocity, was specifically associated with recall of autobiographical memories from real life rather than performance on the more constrained laboratory-based memory tests.

## Discussion

The conduction velocity of action potentials along axons is crucial for neural communication. Until recently it was not possible to examine metrics associated with axonal conduction velocity, such as the g-ratio, in vivo in the human brain, with this being largely the preserve of studies involving non-human animals. However, by combining diffusion MRI with quantitative structural MRI scans optimised to assess myelination (*Callaghan et al., 2014*), it is now possible to estimate the MR g-ratio in vivo in humans (*Drakesmith et al., 2019*; *Mohammadi et al., 2015*; *Mohammadi and Callaghan, 2021*; *Stikov et al., 2015*). Here, in the first application to human cognition, we found that variations in the MR g-ratio specifically in the parahippocampal cingulum bundle were associated with individual differences in autobiographical memory recall ability in a large sample of healthy adults. Under various modelling assumptions related to MR g-ratio and conduction velocity (detailed below), this was associated with larger inner axon diameters rather than myelin content. Moreover, we identified a possible link between autobiographical memory recall ability and more coherently organised neurites. Our findings were also specific to autobiographical memory retrieval and were not evident for laboratory-based memory tests. These results offer a new perspective on individual differences in autobiographical memory recall ability. This is especially welcome given the current lack of a clear biological explanation for such variations in the healthy population (*Clark et al., 2020*; *Clark et al., 2021a*; *LePort et al., 2012*; *Maguire et al., 2003*; *Palombo et al., 2018*; *Van Petten, 2004*).

A key property of the MR g-ratio is that it provides a non-invasive methodology with which to relate in vivo structural neuroimaging of humans with axonal conduction velocity (*Berman et al., 2019*; *Drakesmith et al., 2019*). Moreover, the identification of a statistically significant relationship with the MR g-ratio can help to guide as to which of inner axon diameter or myelin thickness is more likely to be associated with variations in conduction velocity (*Caeyenberghs et al., 2016*; *Kaller et al., 2017*; *Lakhani et al., 2016*; *Waxman, 1980*; *Xin and Chan, 2020*).

Our inferences rested upon the assumption that a significant relationship with the MR g-ratio reflected faster conduction velocity, given previous work suggesting that faster conduction velocity might be associated with better cognition (e.g. *Brancucci, 2012*; *Dicke and Roth, 2016*; *Miller, 1994*; *Reed and Jensen, 1992*). As shown in *Figure 2*, there are three main scenarios describing how changes in the MR g-ratio are related to the underlying microstructural properties, given a faster conduction velocity. The positive relationship we observed between the parahippocampal cingulum bundle MR g-ratio and autobiographical memory recall ability suggests that this effect was associated predominantly with larger inner axon diameter (*Figure 2A* orange arrow, and *Figure 2D*). By contrast, had a negative correlation between the MR g-ratio and autobiographical memory recall ability been identified, we could instead have inferred that myelination was the relevant microstructural feature. The lack of relationship between parahippocampal cingulum bundle magnetisation transfer saturation values (optimised to assess myelination) and autobiographical memory recall ability provided further corroboration that myelin may not have been influential in this context. Greater myelination is often held to be a prominent influence on behavioural and cognitive performance (*Caeyenberghs et al., 2016*; *Fields and Bukalo, 2020*; *Kaller et al., 2017*; *Lakhani et al., 2016*; *Waxman, 1980*; *Xin and Chan, 2020*). By contrast, our results highlight the potentially important role that the inner axon diameter could be playing in autobiographical memory recall.

We also found that lower neurite dispersion, suggesting more coherent neurite organisation, was related to better autobiographical memory recall ability. Our measure of neurite dispersion was obtained using the NODDI biophysical model (*Zhang et al., 2012*), which aims to isolate the organisation of the neurites in a voxel from the density of the neurites in a voxel. While neurite dispersion was significantly related to autobiographical memory recall, no relationship was observed with neurite density.

A larger inner axon diameter reduces resistance to action potential signals, enabling greater conduction velocities (*Gasser and Grundfest, 1939*; *Hursh, 1939*), and coherently organised fibres may decrease the distance signals need to travel, further reducing communication times (*Salami et al., 2003*; *Steriade, 1995*). This combination of features might optimise a fibre bundle for faster communication, and this may benefit autobiographical memory recall.

These results were specific to one white matter tract, the parahippocampal cingulum bundle (*Bubb et al., 2018*). MR g-ratio measures from no other tract, including the dorsal part of the cingulum bundle, showed any association with autobiographical memory recall ability. The parahippocampal cingulum bundle directly connects the hippocampus, parahippocampal, retrosplenial and parietal cortices, the anterior thalamus and, through other subdivisions of the cingulum bundle, the medial prefrontal cortex. These regions are typically engaged during fMRI studies of autobiographical memory recall (*Andrews-Hanna et al., 2014*; *Maguire, 2001*; *Spreng et al., 2009*; *Svoboda et al., 2006*), and damage to them is often associated with autobiographical memory impairments (*Berryhill et al., 2007*; *McCormick et al., 2018*; *Scoville and Milner, 1957*; *Vann et al., 2009*). The retrosplenial and parahippocampal cortices are thought to provide visuospatial elements of autobiographical memories (*Dalton and Maguire, 2017*; *Epstein and Higgins, 2007*; *Mullally and Maguire, 2011*; *Vann et al., 2009*), while the medial prefrontal cortex may initiate autobiographical retrieval and support schema-guided recall (*Gilboa and Marlatte, 2017*; *McCormick et al., 2018*; *McCormick et al., 2020*). The parahippocampal cingulum bundle is, therefore, uniquely positioned as a transmission highway enabling this information to reach the hippocampus, where memories can be reconstructed (*Bartlett, 1932*; *Hassabis and Maguire, 2007*; *Schacter et al., 2012*). Larger inner axon diameters and coherently organised neurites could facilitate rapid information flow along the parahippocampal cingulum bundle leading to more memory elements being available simultaneously, which in turn may result in increased detail and better integration of a memory representation.

There have only been a small number of previous studies investigating white matter tracts and individual differences in autobiographical memory recall ability in healthy people. These focused on

standard DTI parameters, such as fractional anisotropy and mean diffusivity, and not the MR g-ratio. Two such studies were suggestive of a relationship between parahippocampal cingulum bundle fractional anisotropy and autobiographical memory recall (*Irish et al., 2014*; *Memel et al., 2020*). Our analyses of the standard DTI parameters, therefore, replicated these previous results. Direct comparison between these studies and our findings is difficult, however, given the relatively small sample sizes, older age of participants in some of the extant studies, and the testing of mixed groups of healthy older people and patients with dementia. In addition, fractional anisotropy and other physical parameters lack biological specificity (*Jensen and Helpern, 2010*; *Jones et al., 2013a*) and, consequently, they cannot speak to questions concerning axonal conduction velocity, which is our main interest here. In a similar vein, we are not aware of any reports of bilateral lesions that selectively compromise or sever the parahippocampal cingulum bundle in humans. Given our findings, we would predict that such lesions would adversely affect the ability to recall autobiographical memories.

Despite its prime location and connectivity at the heart of the brain's autobiographical memory system, the dearth of studies in humans and the rarity of selective bilateral lesions to the parahippocampal cingulum bundle have perhaps obscured its importance when compared to other, more celebrated, memory-related white matter tracts. We also examined two such tracts, the fornix and the uncinate fasciculus, but in both cases no significant relationships between any of our neuroimaging metrics and autobiographical memory recall ability were evident in our large cohort of young healthy adults. As outlined in *Figure 2*, there was one possible scenario where a null relationship with the MR g-ratio could reflect an association with conduction velocity – if both the inner axon diameter and myelin thickness increased proportionally to each other (*Figure 2A* black arrow, and *Figure 2E*). However, we found no associations between autobiographical memory recall and both the MR g-ratio *and* magnetisation transfer saturation values of the fornix and uncinate fasciculus, which speaks against this explanation. In terms of standard DTI parameters, unlike a previous study (*Hodgetts et al., 2017*), we did not find a relationship between fornix fractional anisotropy and autobiographical memory recall ability, which may be due to the larger sample that we examined.

Both the fornix and parahippocampal cingulum bundle are vulnerable to partial volume effects (*Concha et al., 2005*). However, to mitigate this issue we took steps to ensure the data were extracted only from white matter voxels (see Materials and methods). Diffusion data are also susceptible to distortions which can particularly affect the uncinate fasciculus. We addressed this challenge by using a new technique that improves distortion correction in this region (*Clark et al., 2021b*). The absence of fornix findings in our study could echo those relating to hippocampal volume (*Clark et al., 2020*; *Clark et al., 2021a*), whereby the structure is widely acknowledged to be involved in autobiographical memory recall, and damage impedes retrieval (*Aggleton et al., 2000*; *D'Esposito et al., 1995*; *Gaffan and Gaffan, 1991*; *Tsivilis et al., 2008*), but microstructural variations have limited impact within the healthy adult population. Regarding the uncinate fasciculus, unilateral lesions do not seem to significantly impair performance on laboratory-based memory tests (*Papagno et al., 2011*) or the recall of premorbid autobiographical memories (*Levine et al., 2009*). Bilateral uncinate fasciculus lesions are very rare in humans, but might result in greater memory impairment. Alternatively, hippocampal-prefrontal connections may be better served by other pathways, for example, via the fornix or parts of the cingulum bundle.

Our findings were not only specific to the parahippocampal cingulum bundle, but also to the internal details of autobiographical memories, which reflect the episodicity of past experiences. By contrast, our control measure of external details, also from the same task but which concerns non-episodic information in the autobiographical memories, did not correlate with any white matter microstructure metrics. Moreover, no relationships were identified with any of the eight laboratory-based memory tasks examined, highlighting the specific nature of the relationship between parahippocampal cingulum bundle MR g-ratio and, by inference, conduction velocity, and autobiographical memory recall ability. It may be that vivid, detailed, multimodal, autobiographical memories rely on inter-regional connectivity, particularly that supported by the parahippocampal cingulum bundle, to a greater degree than simpler, more constrained laboratory-based memory tests. This result aligns with previous work involving an fMRI meta-analysis which showed that recall of autobiographical memories and laboratory-based memory stimuli were associated with substantially different neural substrates (*McDermott et al., 2009*). More generally, our findings add to the increasingly-recognised importance of studying real-world cognition in order to fully characterise brain-behaviour relationships

(*Maguire, 2001*; *Maguire, 2012*; *Miller et al., 2022*; *Mobbs et al., 2021*; *Nastase et al., 2020*; *Spiers and Maguire, 2007*).

Age, gender, scanner and the number of voxels within an ROI were included as covariates in all analyses, limiting the potential confounding effects of these variables. Our analyses were also performed using weighted means from each of the three white matter tracts of interest rather than voxel-wise across the whole brain, reducing the potential for false positives (*Marek et al., 2022*). We also tested a sample of over 200 participants, which has been suggested as sufficient for correlational neuroimaging research such as that performed here (*Cecchetti and Handjaras, 2022*; *DeYoung et al., 2022*). Relationships with autobiographical memory recall ability were also specific to the MR g-ratio, and were not evident for any of its components (magnetisation transfer saturation, neurite density or isotropic volume fraction), suggesting that it is the combination of these measures, in the form of the MR g-ratio, that is meaningful rather than any single measure alone.

Nevertheless, as with all neuroimaging techniques, methodological limitations need to be considered when measuring the MR g-ratio (see *Campbell et al., 2018*; *Mohammadi and Callaghan, 2021* for in depth methodological reviews). First, the MR g-ratio is an area-weighted average of all microscopic axons in an MRI-voxel that is slightly weighted towards larger axons. Second, MR proxies are required to estimate myelin and axonal volumes for the calculation of the MR g-ratio. Multiple methodologies are available to estimate myelin and axonal volumes, with no consensus yet reached as to the best combination (*Mohammadi and Callaghan, 2021*). Third, a calibration step is required to more closely align the MR proxies with the estimated volume fractions. Here, we used the standard single-point calibration method to estimate the slope of the myelin-based proxy, assuming that the offset can be neglected. Future work may be able to improve these calibrations. But, of note, since a non-negligible offset in the myelin-based proxy can increase the error in the MR g-ratio, any calibration improvements would likely serve to increase the observed correlation. Fourth, although the model of the MR g-ratio allows for fibre dispersion (*Stikov et al., 2015*), this is not accounted for in mapping to conduction velocity. Going forward, more advanced models may help to further elaborate on these relationships. Fifth, our myelin measure can be influenced by any factor, cognitive or demographic, that leads to a difference in absolute myelination. In principle, unexplained variance could obscure a true underlying relationship with autobiographical memory. To mitigate this, our analyses controlled for age, gender and any potential scanner-related differences. The MR g-ratio, however, is less affected by this limitation because it is a relative measure and does not depend on absolute myelin content but on the balance between absolute myelin and axonal volumes.

Finally, our inference suggesting that the positive correlation between the MR g-ratio and autobiographical memory recall ability is potentially related to inner axon diameter is specifically in the context of g-ratio and conduction velocity, and is based upon Rushton's (1951) model rather than direct measurements of axon diameter. The use of Connectom MRI scanners (*Jones et al., 2018*) and more advanced biophysical models may be able to expand upon this relationship in the future.

While we examined the parahippocampal cingulum bundle as a unitary pathway, it comprises both long and short association fibres with differing connectivity (*Bubb et al., 2018*). Some fibres will form long range connections between, for example, the hippocampus and retrosplenial cortex, whereas others will make shorter range connections between neighbouring regions. Connectom MRI is starting to make examination of short range 'u-fibres' possible in vivo in humans (*Movahedian Attar et al., 2020*; *Shastin et al., 2022*). While there are currently only four of these scanners in the world, future studies using Connectom MRI could seek to identify specific connections *within* the parahippocampal cingulum bundle that relate to individual differences in autobiographical memory recall.

In conclusion, white matter microstructure measures related to conduction velocity are now possible to derive in vivo in humans. This has the potential to provide novel insights into how the brain processes and integrates information (*Berman et al., 2019*; *Drakesmith et al., 2019*), deepening our understanding of the information flow that underpins critical cognitive functions such as autobiographical memory recall.

# Materials and methods

## Participants

Two hundred and seventeen healthy people took part in the study, including 109 females and 108 males. The age range was restricted to 20–41 years old to limit the possible effects of aging (mean age = 29.0 years, SD = 5.60). Participants had English as their first language and reported no history of psychological, psychiatric or neurological conditions. Our aim was to assess people from the general population who would not be classed as having extreme expertise on classic hippocampal tasks, as this could affect hippocampal structure (*Maguire et al., 2000*; *Woollett and Maguire, 2011*). Consequently, people with vocations such as taxi driving (or those training to be taxi drivers), ship navigators, aeroplane pilots, or those with regular hobbies including orienteering, or taking part in memory sports and competitions, were excluded. Of the approximately 2000 people who contacted us, 23 were explicitly excluded on this basis. Participants were reimbursed £10 per hour for taking part which was paid at study completion. All participants gave written informed consent and the study was approved by the University College London Research Ethics Committee (project ID: 6743/001).

A sample size of 217 was determined during study design to be robust to employing different statistical approaches when answering multiple questions of interest. Specifically, the sample allowed for sufficient power to identify medium effect sizes when conducting correlation analyses at alpha levels of 0.01 and when comparing correlations at alpha levels of 0.05 (*Cohen, 1992*). Samples of over 200 participants have also been suggested as sufficient for correlational neuroimaging research similar to that performed here (*Cecchetti and Handjaras, 2022*; *DeYoung et al., 2022*).

## Procedure

Participants completed the study over multiple visits. Diffusion imaging and magnetisation transfer saturation scans were acquired on two separate days, and the Autobiographical Interview was conducted during a third visit. All participants completed all parts of the study.

## The autobiographical interview

This widely-used test (*Levine et al., 2002*) was employed to measure autobiographical memory recall ability. Participants are asked to provide autobiographical memories from a specific time and place over four time periods – early childhood (up to age 11), teenage years (aged from 11 to 17), adulthood (from age 18 years to 12 months prior to the interview; two memories are requested), and the last year (a memory from the last 12 months); therefore, five memories in total are harvested. Recordings of the memory descriptions are transcribed for later scoring.

The main outcome measure of the Autobiographical Interview is the mean number of internal details included in the description of an event from across the five autobiographical memories. Internal details are those describing the event in question (i.e. episodic details) and include event, place, time and perceptual information, as well as thoughts and emotions relating to the event itself. We used the secondary outcome measure of the Autobiographical Interview, the mean number of external details included in the five autobiographical memories, as a control measure. External details include semantic information concerning the event, or other non-event information, and are not considered to reflect autobiographical memory recall ability.

Double scoring was performed on 20% of the data. Inter-class correlation coefficients, with a two-way random effects model looking for absolute agreement were calculated for both internal and external details. This was performed both for individual memories and as an average of all five memories across each participant. For internal details the coefficients were 0.94 and 0.97, respectively, and for external details they were 0.84 and 0.87 respectively. For reference, a score of 0.8 or above is considered excellent agreement beyond chance.

## Laboratory-based memory tests

Eight laboratory-based memory tasks were also administered to participants. These are standard memory tests that are often used in neuropsychological settings. Tasks were performed and scored in line with their standardised and published protocols.

The ability to recall a short narrative was examined using the immediate and delayed recall tests of the Logical Memory subtest of the Wechsler Memory Scale IV (*Wechsler, 2009*). Verbal list recall was assessed using the immediate and delayed recall tests of the Rey Auditory Verbal Learning Test (see

*Strauss et al., 2006*). Visuospatial recall was examined using the delayed recall of the Rey–Osterrieth Complex Figure (*Rey, 1941*). Recognition memory was investigated using the Warrington Recognition Memory Tests for Words and Faces (*Warrington, 1984*). Finally, participants also underwent the 'Dead or Alive' task which probes general knowledge about whether famous individuals have died or are still alive, providing a measure of semantic memory (*Kapur et al., 1989*).

## Diffusion MRI data acquisition

Three MRI scanners were used to collect the neuroimaging data. All scanners were Siemens Magnetom TIM Trio systems with 32 channel head coils and were located at the same neuroimaging centre, running the same software. The sequences were loaded identically onto the individual scanners. Participant set-up and positioning followed the same protocol for each scanner.

Diffusion-weighted images were collected using the multiband accelerated EPI pulse sequence developed by the Centre for Magnetic Resonance Research at the University of Minnesota (R012a-c, R013a on VB17, https://www.cmrr.umn.edu/multiband/; *Feinberg et al., 2010*; *Xu et al., 2013*). Acquisition parameters were: resolution = 1.7 mm isotropic; FOV = 220 mm × 220 mm × 138 mm; 60 directions with 6 interleaved b0 images, echo time (TE)=112ms, repetition time (TR)=4.84 s, with a multiband acceleration factor of 3. The sequence was performed 4 times – twice with b-values of 1000 and twice with b-values of 2500. The first acquisition of each set of b-values was performed with phase-encoding in the anterior to posterior direction (blip-up), the second in the posterior to anterior direction (blip-down). The total acquisition time was 22 min.

## Magnetisation transfer saturation data acquisition

The specific scanner used to collect a participant's diffusion-weighted images was also used to obtain their magnetisation transfer saturation map.

Whole brain structural maps of magnetisation transfer saturation, at an isotropic resolution of 800 μm, were derived from a multi-parameter mapping quantitative imaging protocol (*Callaghan et al., 2015*; *Callaghan et al., 2019*; *Weiskopf et al., 2013*). This protocol consisted of the acquisition of three multi-echo gradient-echo acquisitions with either proton density, T1 or magnetisation transfer weighting. Each acquisition had a TR of 25ms. Proton density weighting was achieved with an excitation flip angle of $6^0$, which was increased to $21^0$ to achieve T1 weighting. Magnetisation transfer weighting was achieved through the application of a Gaussian RF pulse 2 kHz off resonance with 4ms duration and a nominal flip angle of $220^0$. This acquisition had an excitation flip angle of $6^0$. The field of view was 256 mm head-foot, 224 mm anterior-posterior, and 179 mm right-left. The multiple gradient echoes per contrast were acquired with alternating readout gradient polarity at eight equidistant echo times ranging from 2.34 to 18.44ms in steps of 2.30ms using a readout bandwidth of 488 Hz/pixel. Only six echoes were acquired for the magnetisation transfer weighted volume to facilitate the off-resonance pre-saturation pulse and subsequent spoiling gradient within the TR. To accelerate the data acquisition, partially parallel imaging using the GRAPPA algorithm was employed in each phase-encoded direction (anterior-posterior and right-left) with forty integrated reference lines and a speed up factor of two. Calibration data were also acquired at the outset of each session to correct for inhomogeneities in the RF transmit field (*Lutti et al., 2010*). The total acquisition time was 27 min.

## Diffusion MRI pre-processing

The diffusion MRI data were processed using the ACID toolbox (https://www.diffusiontools.com) within SPM12 (https://www.fil.ion.ucl.ac.uk/spm). The weighted average consecutive HySCO pipeline described in *Clark et al., 2021b* was followed, with the addition of multi-shell Position-Orientation Adaptive Smoothing (msPOAS; *Becker et al., 2012*) and Rician bias correction (*Andrews-Hanna et al., 2014*). In brief, the blip-up and blip-down data were first separately corrected for motion and eddy current artefacts. Next, msPOAS was performed, followed by correction for susceptibility-related distortion artefacts using the HySCO2 module (*Macdonald and Ruthotto, 2018*; *Ruthotto et al., 2012*). Tensor fitting (*Mohammadi et al., 2013*) was then implemented separately on each of the distortion corrected blip-up and blip-down datasets to estimate FA maps. HySCO2 was then repeated using the distortion corrected and brain-masked FA maps as input instead of b0 images; the second HySCO2 field map being consecutively applied to the 'pre-corrected' diffusion MRI data. Finally, Rician bias noise correction was employed on the distortion corrected data (*André et al.,*

*2014*), before the data were combined using a weighted average to minimise information loss due to susceptibility distortion blurring induced by local spatial compression.

## Magnetisation transfer saturation pre-processing

The magnetisation transfer saturation data were processed for each participant using the hMRI toolbox (*Tabelow et al., 2019*) within SPM12. The default toolbox configuration settings were used, with the exception that correction for imperfect spoiling was additionally enabled (*Corbin and Callaghan, 2021*). The output magnetisation transfer saturation map quantified the degree of saturation of the steady state signal induced by the application of the off-resonance pre-pulse, having accounted for spatially varying T1 times and RF field inhomogeneity (*Weiskopf et al., 2013*).

Each participant's magnetisation transfer saturation map was segmented into white matter probability maps using the unified segmentation approach (*Ashburner and Friston, 2005*), but with no bias field correction (since the magnetisation transfer saturation map does not suffer from any bias field modulation) and using the tissue probability maps developed by *Lorio et al., 2016*.

## Diffusion model fitting

The MR g-ratio was calculated according to *Ellerbrock and Mohammadi, 2018*:

$$g_{MR} = \sqrt{1 - \frac{MVF_{MR}}{MVF_{MR} + AVF_{MR}}}$$

with $MVF_{MR}$ being the myelin-volume fraction estimated from the magnetisation transfer saturation map and $AVF_{MR}$ being the axonal-volume fraction. The $AVF_{MR}$ was estimated as $AVF_{MR} = (1 - MVF_{MR})$ AWF according to *Stikov et al., 2015*, where AWF was obtained by combining the intra-cellular fraction ($\nu_{icvf}$) and isotropic fraction ($\nu_{iso}$) maps from NODDI (*Zhang et al., 2012*) as $AWF = (1 - \nu_{iso}) \nu_{icvf}$. The magnetisation transfer saturation map was obtained from the hMRI toolbox as described above. For calibration of the magnetisation transfer saturation map to a myelin-volume fraction map $(MVF_{MR} = \alpha\, MT_{sat})$, we used the g-ratio based calibration method as reported in *Ellerbrock and Mohammadi, 2018* and *Mohammadi and Callaghan, 2021*, with $\alpha$=0.1683.

The NODDI biophysical model (*Zhang et al., 2012*) was also used to obtain maps of the neurite orientation dispersion index and neurite density using the NODDI toolbox (http://mig.cs.ucl.ac.uk/index.php?n=Tutorial.NODDImatlab).

Finally, for completeness, Axial-Symmetric DKI (*Oeschger et al., 2021*) was performed on the pre-processed diffusion data using the ACID toolbox to generate maps of the more commonly reported physical parameters of fractional anisotropy, mean diffusivity, mean kurtosis, diffusivities parallel and diffusivities perpendicular.

## Microstructure data extraction

Microstructure data extraction was performed in Montreal Neurological Institute (MNI) space. The diffusion and magnetisation transfer saturation maps were transformed from native to MNI space using the hMRI toolbox (*Tabelow et al., 2019*). This involved performing inter-subject registration using DARTEL (*Ashburner, 2007*) on the segmented magnetisation transfer saturation grey and white matter probability maps, with the resulting DARTEL template and deformations then used to normalize the diffusion and magnetisation transfer saturation maps to MNI space at 1.5 x 1.5 x 1.5mm.

Bilateral tract ROIs were defined using the Johns Hopkins probabilistic white matter tractography atlas (*Hua et al., 2008*), with the exception of the fornix which was defined using the ICBM-DTI-81 white-matter labels atlas (*Mori et al., 2008*) as the fornix is not available in the probabilistic atlas. Our primary foci were the fornix, uncinate fasciculus and parahippocampal cingulum bundle. However, we also performed exploratory analyses on six other tracts - the anterior thalamic radiation, the dorsal cingulum bundle, forceps minor, inferior longitudinal fasciculus, inferior occipitofrontal fasciculus and superior longitudinal fasciculus. To reduce partial volume effects, for all tracts (with the exception of the fornix as the available fornix tract was not probabilistic) the minimum probability threshold was set to 25%. In addition, all of the tract ROIs were refined for each participant using their segmented magnetisation transfer saturation white matter probability map, with a minimum probability of 90% to limit the mask to white matter. This also served to remove any residual mis-alignment from the maps

being transformed into MNI space, as no smoothing was performed to preserve the quantitative values. As this resulted in differing tract ROI sizes for each participant, the number of voxels in each tract for each participant was calculated. Mean values of the extracted microstructure metrics from the tract ROIs were determined using a weighted average, where voxels with higher white matter probabilities contributed more to the mean.

We used the well-established ROI approach (e.g. *Ellerbrock and Mohammadi, 2018*; *Memel et al., 2020*) rather than, for example, a tract-based pipeline, in order to reduce the influence of seed and target region selection. This is because small changes to these selections can result in different tracts being identified. We also wanted to avoid the inclusion of excess grey matter in the tracts themselves, because it is not possible to estimate the MR g-ratio in grey matter tissue. In addition, an ROI approach reduced the potential for false positives in comparison to performing tract-based voxel-wise analyses.

## Statistical analyses

Analyses were performed in SPSS v27 unless otherwise stated. Data were summarised using means and standard deviations. There were no missing data, and no data needed to be removed from any analysis.

As we had different tract ROI sizes for each participant, we first assessed whether there were any relationships between the number of voxels in the tract ROIs and autobiographical memory recall ability. We performed partial correlations for each tract between the number of voxels in the tract ROI and the number of internal details on the Autobiographical Interview, with age, gender and scanner as covariates. No significant relationships were identified (all $r < 0.12$, all $p > 0.1$). However, to ensure no residual effects were present, the number of voxels in a tract ROI was included as a covariate in the analyses.

In our main analyses, we first investigated the relationships between each microstructure measure and the number of internal details from the Autobiographical Interview using partial correlations, with bootstrapping performed 10,000 times to calculate confidence intervals. Four covariates were included in each partial correlation: age, gender, scanner, and the number of voxels in a tract ROI. For these primary analyses, similar partial correlations were performed for the external details control measure. If an internal details correlation was significant, the internal and external details correlations were then directly compared in order to test for statistical difference using the technique described by *Meng et al., 1992*. This approach extends the Fisher z transformation, allowing for more accurate testing and comparison of two related correlations. The correlation comparison was performed using the R cocor package v1.1.3 (*Diedenhofen and Musch, 2015*).

We also investigated the relationship between the MR g-ratio and the magnetisation transfer saturation values and eight laboratory-based memory tests. As with the main analyses, partial correlations were performed between the MR g-ratio and magnetisation transfer saturation values and the outcome measures of the memory tests with age, gender, scanner and the number of voxels in a tract ROI included as covariates, and bootstrapping performed 10,000 times to calculate confidence intervals.

As the microstructure measures were investigated across several tracts, we corrected for the repeated testing of the same measures across our three main tracts of interest (the fornix, the uncinate fasciculus and the parahippocampal cingulum bundle) using the Bonferroni method; dividing alpha = 0.05 by 3. Consequently, associations with a two-sided p-value <0.017 were considered significant. As the comparison of correlations was performed only when a significant correlation was identified, a two-sided p-value <0.05 was deemed significant.

## Acknowledgements

Thanks to Anna Monk, Victoria Hotchin, Gloria Pizzamiglio and Alice Liefgreen for assistance with data collection and scoring. Thanks also to Mohammad Ashtarayeh and Jan Malte Oeschger for their support with implementing the diffusion model fitting and g-ratio calculations.

## Additional information

### Funding

| Funder | Grant reference number | Author |
|---|---|---|
| Wellcome Trust | 101759/Z/13/Z | Eleanor A Maguire |
| Wellcome Trust | 210567/Z/18/Z | Eleanor A Maguire |
| Wellcome Trust | 203147/Z/16/Z | Eleanor A Maguire |
| ERA-NET NEURON | hMRI-ofSCI | Siawoosh Mohammadi Martina F Callaghan |
| Federal Ministry of Education and Research | 01EW1711A and B | Siawoosh Mohammadi |
| German Research Foundation | MO 2397/5-1 | Siawoosh Mohammadi |
| German Research Foundation | MO 2397/4-1 | Siawoosh Mohammadi |
| Forschungszentrums Medizintechnik Hamburg | 01fmthh2017 | Siawoosh Mohammadi |
| MRC and Spinal Research Charity | MR/R000050/1 | Martina F Callaghan |

The funders had no role in study design, data collection and interpretation, or the decision to submit the work for publication. For the purpose of Open Access, the authors have applied a CC BY public copyright license to any Author Accepted Manuscript version arising from this submission.

### Author contributions

Ian A Clark, Conceptualization, Formal analysis, Investigation, Methodology, Writing - original draft, Writing - review and editing; Siawoosh Mohammadi, Martina F Callaghan, Formal analysis, Writing - review and editing; Eleanor A Maguire, Conceptualization, Formal analysis, Supervision, Funding acquisition, Methodology, Writing - original draft, Writing - review and editing

### Author ORCIDs

Ian A Clark http://orcid.org/0000-0002-5678-2190
Siawoosh Mohammadi http://orcid.org/0000-0003-1311-9636
Eleanor A Maguire http://orcid.org/0000-0002-9470-6324

### Ethics

All participants gave written informed consent, including consent to publish, and the study was approved by the University College London Research Ethics Committee (project ID: 6743/001).

### Decision letter and Author response

Decision letter https://doi.org/10.7554/eLife.79303.sa1
Author response https://doi.org/10.7554/eLife.79303.sa2

## Additional files

### Supplementary files

• Supplementary file 1. Microstructure and standard DTI parameters data for the fornix and uncinate fasciculus.

• Supplementary file 2. Microstructure and standard DTI parameters data for the parahippocampal cingulum bundle and source data for *Appendix 1—figures 9 and 10*.

• MDAR checklist

### Data availability

The data for every participant are provided in Figure 5—source data 1, Figure 6—source data 1, Supplementary file 1 and Supplementary file 2.

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

# Appendix 1

## Details of the simulation performed for *Figure 2A*

In *Figure 2A*, inner axon diameter, myelin thickness and g-ratio are plotted together. The range of g-ratio values used in the simulation spanned two standard deviations about the mean MR g-ratio observed for the parahippocampal cingulum bundle in the current study (mean = 0.647, standard deviation = 0.043). The axon diameter was computed for this MR g-ratio range by re-arranging the equation presented in *Berman et al., 2019* such that axon diameter $= \exp\left(\frac{\text{g-ratio}-0.506}{0.22}\right)$. Fibre dimeter was then calculated as fibre diameter $= \frac{\text{axon diamter}}{\text{g-ratio}}$, enabling myelin thickness to be computed as myelin thickness $= \frac{\text{fibre diameter}-\text{axon diameter}}{2}$.

We note that discrepancies between reported microscopic parameters (i.e. g-ratio, modelled axon diameter and myelin thickness) derived from in vivo and ex vivo histology may arise for two reasons. (1) The in vivo MR g-ratio is computed from volume-fractions unlike the microscopic g-ratio measured with histology. (2) The heuristic equation by *Berman et al., 2019* that is relating the in vivo MR g-ratio to axon diameter is rather capturing the tail of the axon radii distribution.

## Investigation of the MR g-ratio, magnetisation transfer saturation values, neurite orientation dispersion index, neurite density and standard DTI parameters in the fornix and uncinate fasciculus

As can be observed in the tables below, there were no significant correlations between microstructural measures or standard DTI parameters and autobiographical memory recall ability for either of the tracts when using the corrected (p<0.017) threshold.

### Fornix

The mean number of voxels in the region of interest (ROI) was 249.55 (SD = 23.09).

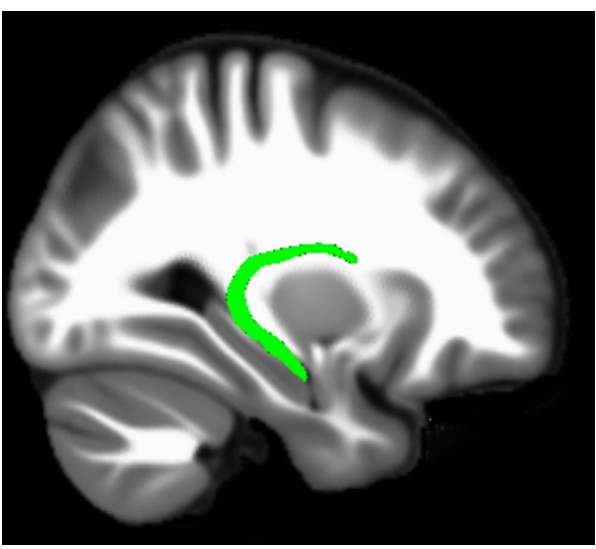

**Appendix 1—figure 1.** The location of the fornix.

**Appendix 1—table 1.** Means and standard deviations for the microstructure measures and standard DTI parameters extracted from the fornix.

| Measure | Mean | Standard deviation |
|---|---|---|
| MR g-ratio | 0.720 | 0.017 |

*Appendix 1—table 1 Continued on next page*

*Appendix 1—table 1 Continued*

| Measure | Mean | Standard deviation |
|---|---|---|
| Magnetisation transfer saturation | 0.973 | 0.003 |
| Neurite dispersion (ODI) | 0.143 | 0.021 |
| Neurite density | 0.601 | 0.045 |
| Fractional anisotropy | 0.605 | 0.035 |
| Mean diffusivity ($10^{-3}$ mm²/s) | 0.901 | 0.036 |
| Mean kurtosis | 0.909 | 0.100 |
| Diffusivities parallel ($10^{-3}$ mm²/s) | 1.645 | 0.072 |
| Diffusivities perpendicular ($10^{-3}$ mm²/s) | 0.529 | 0.042 |

Note. ODI = Orientation Dispersion Index.

**Appendix 1—table 2.** Partial correlations between the microstructure measures or standard DTI parameters extracted from the fornix and autobiographical memory recall ability (internal details).

| Measure | r(211) | p | 95% Confidence interval | |
|---|---|---|---|---|
| | | | Lower | Upper |
| MR g-ratio | –0.04 | 0.53 | –0.16 | 0.08 |
| Magnetisation transfer saturation | –0.08 | 0.23 | –0.20 | 0.03 |
| Neurite dispersion (ODI) | –0.03 | 0.62 | –0.16 | 0.10 |
| Neurite density | –0.03 | 0.69 | –0.15 | 0.10 |
| Fractional anisotropy | 0.02 | 0.80 | –0.12 | 0.15 |
| Mean diffusivity | 0.07 | 0.35 | –0.08 | 0.21 |
| Mean kurtosis | –0.03 | 0.66 | –0.16 | 0.10 |
| Diffusivities parallel | 0.05 | 0.46 | –0.09 | 0.19 |
| Diffusivities perpendicular | 0.02 | 0.74 | –0.12 | 0.15 |

Note. ODI = Orientation Dispersion Index.

## Uncinate fasciculus

The mean number of voxels in the ROI was 191.98 (SD = 29.77).

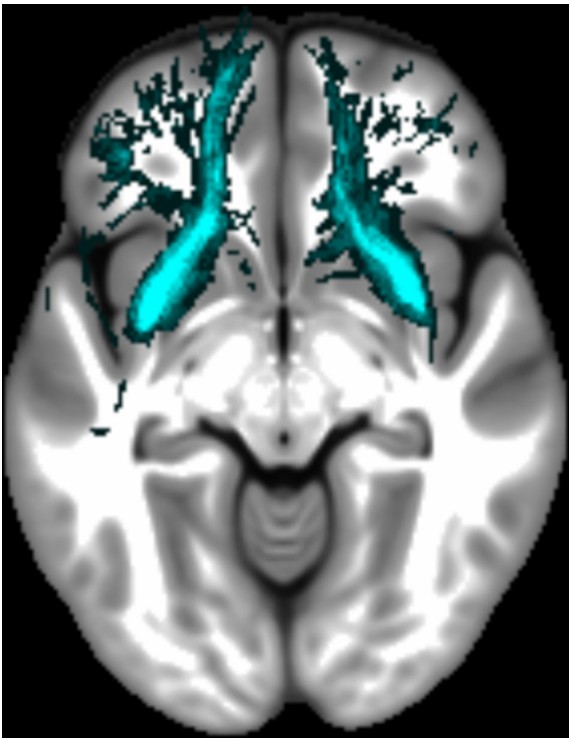

**Appendix 1—figure 2.** The location of the uncinate fasciculus.

**Appendix 1—table 3.** Means and standard deviations for the microstructure measures and standard DTI parameters extracted from the uncinate fasciculus.

| Measure | Mean | Standard deviation |
| --- | --- | --- |
| MR g-ratio | 0.723 | 0.016 |
| Magnetisation transfer saturation | 0.972 | 0.003 |
| Neurite dispersion (ODI) | 0.189 | 0.022 |
| Neurite density | 0.562 | 0.045 |
| Fractional anisotropy | 0.512 | 0.036 |
| Mean diffusivity ($10^{-3}$ mm$^2$/s) | 0.877 | 0.304 |
| Mean kurtosis | 0.913 | 0.100 |
| Diffusivities parallel ($10^{-3}$ mm$^2$/s) | 1.456 | 0.055 |
| Diffusivities perpendicular ($10^{-3}$ mm$^2$/s) | 0.588 | 0.039 |

Note. ODI = Orientation Dispersion Index.

**Appendix 1—table 4.** Partial correlations between the microstructure measures or standard DTI parameters extracted from the uncinate fasciculus and autobiographical memory recall ability (internal details).

| Measure | r(211) | p | 95% Confidence interval | |
|---|---|---|---|---|
| | | | Lower | Upper |
| MR g-ratio | 0.10 | 0.15 | –0.03 | 0.22 |
| Magnetisation transfer saturation | –0.02 | 0.81 | –0.14 | 0.11 |
| Neurite dispersion (ODI) | –0.01 | 0.94 | –0.14 | 0.13 |
| Neurite density | 0.01 | 0.89 | –0.14 | 0.16 |
| Fractional anisotropy | –0.01 | 0.94 | –0.14 | 0.13 |
| Mean diffusivity | –0.01 | 0.87 | –0.14 | 0.13 |
| Mean kurtosis | –0.01 | 0.91 | –0.13 | 0.18 |
| Diffusivities parallel | –0.01 | 0.87 | –0.14 | 0.13 |
| Diffusivities perpendicular | 0.00 | 0.96 | –0.15 | 0.14 |

Note. ODI = Orientation Dispersion Index.

## Investigation of associations between the MR g-ratio and magnetisation transfer saturation values of the fornix and uncinate fasciculus and the laboratory-based memory tests

As can be observed in the tables below, there were no significant correlations between either the MR g-ratio or magnetisation transfer saturation values of the fornix or uncinate fasciculus and any of the laboratory-based memory tasks for either of the tracts, even when using a p<0.05 uncorrected threshold.

**Appendix 1—table 5.** Partial correlations between the fornix MR g-ratio and magnetisation transfer saturation values and the laboratory-based memory tests.

| Measure | r(211) | p | 95% Confidence interval | |
|---|---|---|---|---|
| | | | Lower | Upper |
| **MR g-ratio** | | | | |
| Logical Memory immediate recall | –0.01 | 0.85 | –0.14 | 0.11 |
| Logical Memory delayed recall | –0.07 | 0.33 | –0.19 | 0.06 |
| RAVLT immediate recall | –0.06 | 0.42 | –0.20 | 0.09 |
| RAVLT delayed recall | 0.00 | 0.98 | –0.14 | 0.15 |
| Rey-Osterrieth Complex Figure delayed recall | 0.10 | 0.16 | –0.04 | 0.23 |
| Warrington RMT for Words | 0.04 | 0.59 | –0.08 | 0.16 |
| Warrington RMT for Faces | –0.05 | 0.52 | –0.17 | 0.09 |
| Dead or Alive Test | –0.05 | 0.50 | –0.16 | 0.09 |

*Appendix 1—table 5 Continued on next page*

*Appendix 1—table 5 Continued*

| Measure | r(211) | p | 95% Confidence interval | |
| --- | --- | --- | --- | --- |
| | | | Lower | Upper |
| **Magnetisation transfer saturation** | | | | |
| Logical Memory immediate recall | 0.04 | 0.53 | –0.11 | 0.19 |
| Logical Memory delayed recall | 0.11 | 0.11 | –0.04 | 0.26 |
| RAVLT immediate recall | 0.01 | 0.92 | –0.11 | 0.13 |
| RAVLT delayed recall | 0.03 | 0.71 | –0.09 | 0.15 |
| Rey-Osterrieth Complex Figure delayed recall | 0.07 | 0.33 | –0.06 | 0.19 |
| Warrington RMT for Words | 0.06 | 0.41 | –0.05 | 0.17 |
| Warrington RMT for Faces | 0.04 | 0.61 | –0.09 | 0.16 |
| Dead or Alive Test | 0.08 | 0.25 | –0.06 | 0.21 |

Note. RAVLT = Rey Auditory Verbal Learning Test; RMT = Recognition Memory Test.

**Appendix 1—table 6.** Partial correlations between the uncinate fasciculus MR g-ratio and magnetisation transfer saturation values and the laboratory-based memory tests.

| Measure | r(211) | p | 95% Confidence interval | |
| --- | --- | --- | --- | --- |
| | | | Lower | Upper |
| **MR g-ratio** | | | | |
| Logical Memory immediate recall | 0.07 | 0.34 | –0.06 | 0.19 |
| Logical Memory delayed recall | 0.04 | 0.34 | –0.09 | 0.17 |
| RAVLT immediate recall | 0.00 | 0.98 | –0.14 | 0.13 |
| RAVLT delayed recall | 0.06 | 0.40 | –0.10 | 0.21 |
| Rey-Osterrieth Complex Figure delayed recall | 0.01 | 0.94 | –0.13 | 0.15 |
| Warrington RMT for Words | –0.01 | 0.86 | –0.16 | 0.15 |
| Warrington RMT for Faces | –0.07 | 0.34 | –0.20 | 0.06 |
| Dead or Alive Test | –0.04 | 0.60 | –0.15 | 0.08 |
| **Magnetisation transfer saturation** | | | | |
| Logical Memory immediate recall | –0.04 | 0.53 | –0.17 | 0.09 |
| Logical Memory delayed recall | –0.05 | 0.49 | –0.18 | 0.09 |
| RAVLT immediate recall | –0.13 | 0.06 | –0.26 | 0.01 |
| RAVLT delayed recall | –0.16 | 0.02 | –0.27 | –0.03 |
| Rey-Osterrieth Complex Figure delayed recall | –0.07 | 0.33 | –0.20 | 0.07 |
| Warrington RMT for Words | –0.14 | 0.04 | –0.26 | –0.01 |
| Warrington RMT for Faces | –0.05 | 0.44 | –0.18 | 0.09 |

*Appendix 1—table 6 Continued*

| Measure | r(211) | p | 95% Confidence interval | |
|---|---|---|---|---|
| | | | Lower | Upper |
| Dead or Alive Test | –0.01 | 0.93 | –0.14 | 0.13 |

Note. RAVLT = Rey Auditory Verbal Learning Test; RMT = Recognition Memory Test.

## Exploratory analyses of the MR g-ratio, magnetisation transfer saturation values, neurite orientation dispersion index, neurite density and standard DTI parameters in other white matter tracts

As can be observed in the tables below, there were no significant correlations between microstructural measures or standard DTI parameters and autobiographical memory recall ability for any of the tracts when using the corrected (p<0.017) threshold.

### Anterior thalamic radiation

The mean number of voxels in the ROI was 2090.21 (SD = 67.43).

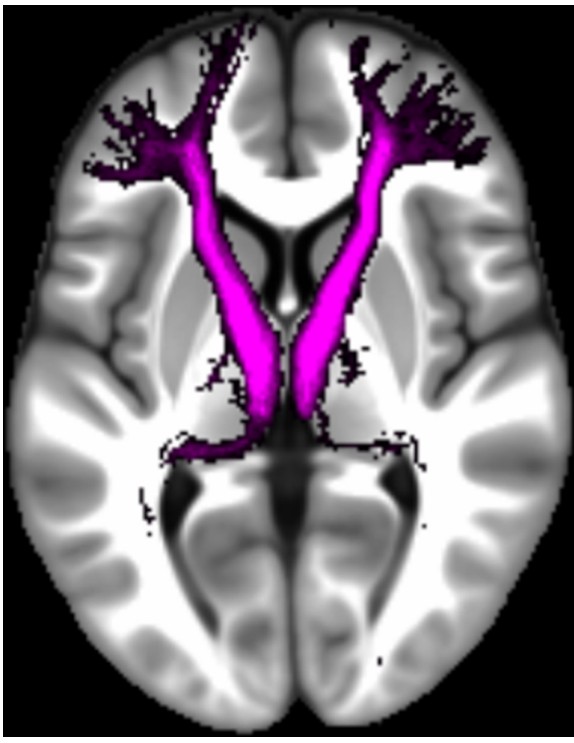

**Appendix 1—figure 3.** The location of the anterior thalamic radiation.

**Appendix 1—table 7.** Means and standard deviations for the microstructure measures and standard DTI parameters extracted from the anterior thalamic radiation.

| Measure | Mean | Standard deviation |
|---|---|---|
| MR g-ratio | 0.724 | 0.014 |
| Magnetisation transfer saturation | 0.986 | 0.001 |

*Appendix 1—table 7 Continued on next page*

*Appendix 1—table 7 Continued*

| Measure | Mean | Standard deviation |
|---|---|---|
| Neurite dispersion (ODI) | 0.252 | 0.017 |
| Neurite density | 0.598 | 0.041 |
| Fractional anisotropy | 0.430 | 0.028 |
| Mean diffusivity ($10^{-3}$ mm$^2$/s) | 0.861 | 0.029 |
| Mean kurtosis | 0.986 | 0.063 |
| Diffusivities parallel ($10^{-3}$ mm$^2$/s) | 1.328 | 0.041 |
| Diffusivities perpendicular ($10^{-3}$ mm$^2$/s) | 0.627 | 0.034 |

Note. ODI = Orientation Dispersion Index.

**Appendix 1—table 8.** Partial correlations between the microstructure measures or standard DTI parameters extracted from the anterior thalamic radiation and autobiographical memory recall ability (internal details).

| Measure | r(211) | p | 95% Confidence interval | |
|---|---|---|---|---|
| | | | Lower | Upper |
| MR g-ratio | 0.05 | 0.47 | −0.06 | 0.17 |
| MT sat | −0.08 | 0.26 | −0.20 | 0.04 |
| Neurite dispersion (ODI) | −0.05 | 0.46 | −0.18 | 0.09 |
| Neurite density | 0.10 | 0.16 | −0.04 | 0.23 |
| Fractional anisotropy | 0.08 | 0.26 | −0.06 | 0.21 |
| Mean diffusivity | 0.03 | 0.67 | −0.11 | 0.17 |
| Mean kurtosis | 0.07 | 0.28 | −0.07 | 0.22 |
| Diffusivities parallel | 0.09 | 0.20 | −0.04 | 0.21 |
| Diffusivities perpendicular | −0.02 | 0.77 | −0.16 | 0.12 |

Note. MT sat = Magnetisation Transfer saturation; ODI = Orientation Dispersion Index.

## Dorsal cingulum bundle
The mean number of voxels in the ROI was 611.48 (SD = 25.29).

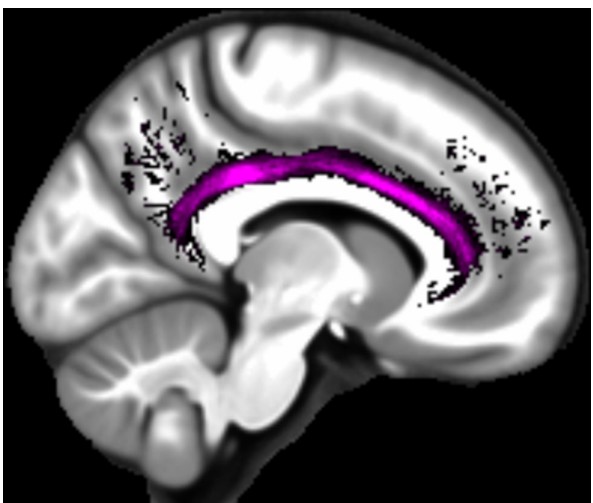

**Appendix 1—figure 4.** The location of the dorsal cingulum bundle.

**Appendix 1—table 9.** Means and standard deviations for the microstructure measures and standard DTI parameters extracted from the dorsal cingulum bundle.

| Measure | Mean | Standard deviation |
|---|---|---|
| MR g-ratio | 0.710 | 0.016 |
| Magnetisation transfer saturation | 0.984 | 0.003 |
| Neurite dispersion (ODI) | 0.147 | 0.019 |
| Neurite density | 0.560 | 0.038 |
| Fractional anisotropy | 0.570 | 0.039 |
| Mean diffusivity ($10^{-3}$ mm$^2$/s) | 0.862 | 0.026 |
| Mean kurtosis | 0.862 | 0.093 |
| Diffusivities parallel ($10^{-3}$ mm$^2$/s) | 1.531 | 0.061 |
| Diffusivities perpendicular ($10^{-3}$ mm$^2$/s) | 0.527 | 0.040 |

Note. ODI = Orientation Dispersion Index.

**Appendix 1—table 10.** Partial correlations between the microstructure measures or standard DTI parameters extracted from the dorsal cingulum bundle and autobiographical memory recall ability (internal details).

| Measure | r(211) | p | 95% Confidence interval Lower | Upper |
|---|---|---|---|---|
| MR g-ratio | 0.09 | 0.22 | −0.04 | 0.21 |
| MT sat | 0.08 | 0.23 | −0.04 | 0.21 |

*Appendix 1—table 10 Continued on next page*

*Appendix 1—table 10 Continued*

| Measure | r(211) | p | 95% Confidence interval | |
|---|---|---|---|---|
| | | | Lower | Upper |
| Neurite dispersion (ODI) | 0.09 | 0.17 | −0.05 | 0.23 |
| Neurite density | 0.09 | 0.17 | −0.05 | 0.23 |
| Fractional anisotropy | 0.01 | 0.85 | −0.13 | 0.16 |
| Mean diffusivity | 0.06 | 0.43 | −0.08 | 0.19 |
| Mean kurtosis | 0.13 | 0.05 | 0.00 | 0.27 |
| Diffusivities parallel | 0.05 | 0.48 | −0.08 | 0.18 |
| Diffusivities perpendicular | 0.01 | 0.91 | −0.14 | 0.15 |

Note. MT sat = Magnetisation Transfer saturation; ODI = Orientation Dispersion Index.

## Forceps minor

The mean number of voxels in the ROI was 4613.67 (SD = 78.02).

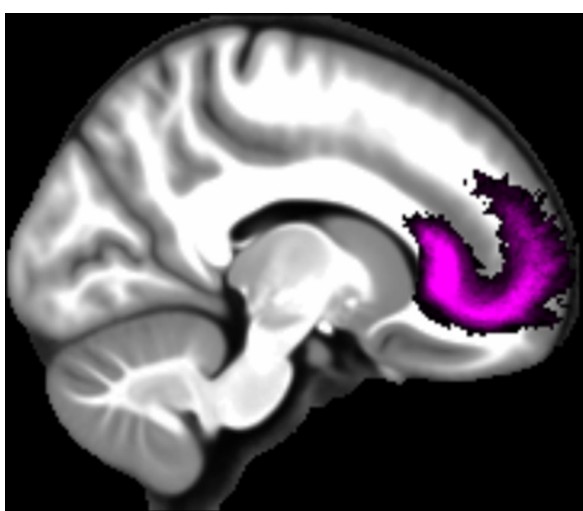

**Appendix 1—figure 5.** The location of the forceps minor.

**Appendix 1—table 11.** Means and standard deviations for the microstructure measures and standard DTI parameters extracted from the forceps minor.

| Measure | Mean | Standard deviation |
|---|---|---|
| MR g-ratio | 0.698 | 0.026 |
| Magnetisation transfer saturation | 0.993 | 0.002 |
| Neurite dispersion (ODI) | 0.201 | 0.016 |

*Appendix 1—table 11 Continued on next page*

*Appendix 1—table 11 Continued*

| Measure | Mean | Standard deviation |
|---|---|---|
| Neurite density | 0.601 | 0.042 |
| Fractional anisotropy | 0.500 | 0.027 |
| Mean diffusivity ($10^{-3}$ mm$^2$/s) | 0.881 | 0.031 |
| Mean kurtosis | 0.947 | 0.128 |
| Diffusivities parallel ($10^{-3}$ mm$^2$/s) | 1.467 | 0.052 |
| Diffusivities perpendicular ($10^{-3}$ mm$^2$/s) | 0.588 | 0.034 |

Note. ODI = Orientation Dispersion Index.

**Appendix 1—table 12.** Partial correlations between the microstructure measures or standard DTI parameters extracted from the forceps minor and autobiographical memory recall ability (internal details).

| Measure | r(211) | p | 95% Confidence interval Lower | Upper |
|---|---|---|---|---|
| MR g-ratio | 0.04 | 0.53 | –0.07 | 0.18 |
| MT sat | 0.07 | 0.34 | –0.06 | 0.19 |
| Neurite dispersion (ODI) | –0.06 | 0.38 | –0.21 | 0.07 |
| Neurite density | 0.09 | 0.19 | –0.04 | 0.22 |
| Fractional anisotropy | 0.10 | 0.17 | –0.03 | 0.23 |
| Mean diffusivity | –0.04 | 0.56 | –0.17 | 0.09 |
| Mean kurtosis | 0.00 | 0.97 | –0.11 | 0.21 |
| Diffusivities parallel | 0.03 | 0.67 | –0.09 | 0.18 |
| Diffusivities perpendicular | –0.07 | 0.32 | –0.19 | 0.05 |

Note. MT sat = Magnetisation Transfer saturation; ODI = Orientation Dispersion Index.

## Inferior longitudinal fasciculus
The mean number of voxels in the ROI was 2844.03 (SD = 52.97).

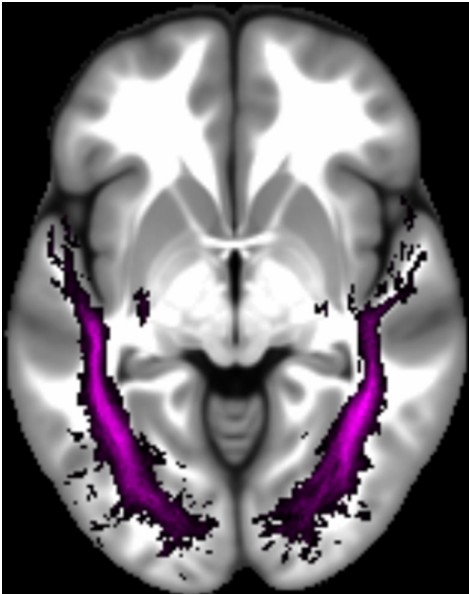

**Appendix 1—figure 6.** The location of the inferior longitudinal fasciculus.

**Appendix 1—table 13.** Means and standard deviations for the microstructure measures and standard DTI parameters extracted from the inferior longitudinal fasciculus.

| Measure | Mean | Standard deviation |
|---|---|---|
| MR g-ratio | 0.724 | 0.013 |
| Magnetisation transfer saturation | 0.994 | 0.002 |
| Neurite dispersion (ODI) | 0.184 | 0.017 |
| Neurite density | 0.558 | 0.041 |
| Fractional anisotropy | 0.486 | 0.028 |
| Mean diffusivity ($10^{-3}$ mm²/s) | 0.902 | 0.029 |
| Mean kurtosis | 0.918 | 0.061 |
| Diffusivities parallel ($10^{-3}$ mm²/s) | 1.473 | 0.045 |
| Diffusivities perpendicular ($10^{-3}$ mm²/s) | 0.616 | 0.035 |

Note. ODI = Orientation Dispersion Index.

**Appendix 1—table 14.** Partial correlations between the microstructure measures or standard DTI parameters extracted from the inferior longitudinal fasciculus and autobiographical memory recall ability (internal details).

| Measure | r(211) | p | 95% Confidence interval | |
|---|---|---|---|---|
| | | | Lower | Upper |

*Appendix 1—table 14 Continued on next page*

*Appendix 1—table 14 Continued*

| Measure | r(211) | p | 95% Confidence interval | |
|---|---|---|---|---|
| | | | Lower | Upper |
| MR g-ratio | 0.07 | 0.35 | –0.06 | 0.19 |
| MT sat | 0.07 | 0.35 | –0.05 | 0.18 |
| Neurite dispersion (ODI) | –0.03 | 0.64 | –0.17 | 0.10 |
| Neurite density | 0.07 | 0.35 | –0.07 | 0.19 |
| Fractional anisotropy | 0.10 | 0.14 | –0.04 | 0.23 |
| Mean diffusivity | –0.01 | 0.87 | –0.15 | 0.13 |
| Mean kurtosis | 0.10 | 0.16 | –0.03 | 0.23 |
| Diffusivities parallel | 0.07 | 0.29 | –0.07 | 0.21 |
| Diffusivities perpendicular | –0.07 | 0.34 | –0.19 | 0.07 |

Note. MT sat = Magnetisation Transfer saturation; ODI = Orientation Dispersion Index.

## Inferior occipitofrontal fasciculus
The mean number of voxels in the ROI was 3344.31 (SD = 46.0).

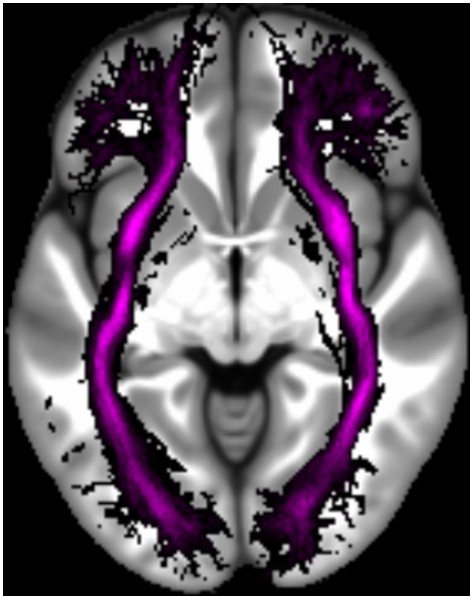

**Appendix 1—figure 7.** The location of the inferior occipitofrontal fasciculus.

**Appendix 1—table 15.** Means and standard deviations for the microstructure measures and standard DTI parameters extracted from the inferior occipitofrontal fasciculus.

| Measure | Mean | Standard deviation |
|---|---|---|
| MR g-ratio | 0.729 | 0.011 |
| Magnetisation transfer saturation | 0.995 | 0.001 |
| Neurite dispersion (ODI) | 0.179 | 0.012 |
| Neurite density | 0.563 | 0.036 |
| Fractional anisotropy | 0.508 | 0.024 |
| Mean diffusivity ($10^{-3}$ mm$^2$/s) | 0.889 | 0.026 |
| Mean kurtosis | 0.912 | 0.057 |
| Diffusivities parallel ($10^{-3}$ mm$^2$/s) | 1.488 | 0.039 |
| Diffusivities perpendicular ($10^{-3}$ mm$^2$/s) | 0.590 | 0.031 |

Note. ODI = Orientation Dispersion Index.

**Appendix 1—table 16.** Partial correlations between the microstructure measures or standard DTI parameters extracted from the inferior occipitofrontal fasciculus and autobiographical memory recall ability (internal details).

| Measure | r(211) | p | 95% Confidence interval Lower | Upper |
|---|---|---|---|---|
| MR g-ratio | –0.01 | 0.93 | –0.12 | 0.12 |
| MT sat | 0.08 | 0.22 | –0.03 | 0.19 |
| Neurite dispersion (ODI) | –0.01 | 0.89 | –0.14 | 0.13 |
| Neurite density | 0.06 | 0.42 | –0.07 | 0.18 |
| Fractional anisotropy | 0.06 | 0.38 | –0.08 | 0.20 |
| Mean diffusivity | 0.00 | 0.99 | –0.13 | 0.13 |
| Mean kurtosis | 0.07 | 0.31 | –0.06 | 0.19 |
| Diffusivities parallel | 0.05 | 0.50 | –0.09 | 0.18 |
| Diffusivities perpendicular | –0.03 | 0.62 | –0.17 | 0.10 |

Note. MT sat = Magnetisation Transfer saturation; ODI = Orientation Dispersion Index.

## Superior longitudinal fasciculus

The mean number of voxels in the ROI was 4243.22 (SD = 39.80).

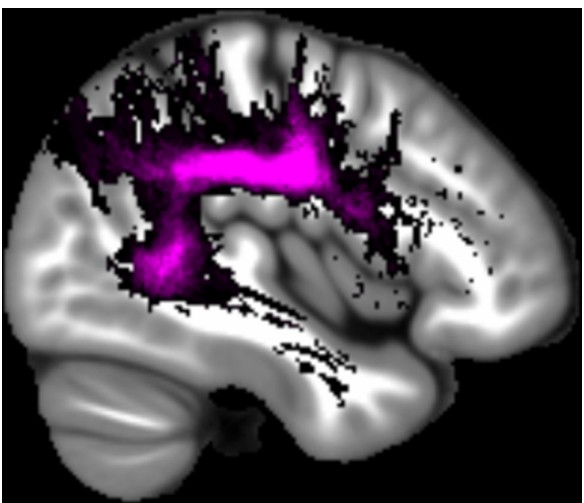

**Appendix 1—figure 8.** The location of the superior longitudinal fasciculus.

**Appendix 1—table 17.** Means and standard deviations for the microstructure measures and standard DTI parameters extracted from the superior longitudinal fasciculus.

| Measure | Mean | Standard deviation |
|---|---|---|
| MR g-ratio | 0.751 | 0.011 |
| Magnetisation transfer saturation | 0.996 | 0.001 |
| Neurite dispersion (ODI) | 0.218 | 0.012 |
| Neurite density | 0.634 | 0.034 |
| Fractional anisotropy | 0.471 | 0.026 |
| Mean diffusivity ($10^{-3}$ mm$^2$/s) | 0.825 | 0.025 |
| Mean kurtosis | 1.029 | 0.041 |
| Diffusivities parallel ($10^{-3}$ mm$^2$/s) | 1.324 | 0.035 |
| Diffusivities perpendicular ($10^{-3}$ mm$^2$/s) | 0.575 | 0.041 |

Note. ODI = Orientation Dispersion Index.

**Appendix 1—table 18.** Partial correlations between microstructure measures or standard DTI parameters extracted from the superior longitudinal fasciculus and autobiographical memory recall ability (internal details).

| Measure | r(211) | p | 95% Confidence interval Lower | Upper |
|---|---|---|---|---|
| MR g-ratio | 0.08 | 0.28 | −0.05 | 0.20 |
| MT sat | −0.04 | 0.54 | −0.16 | 0.08 |

*Appendix 1—table 18 Continued on next page*

*Appendix 1—table 18 Continued*

| Measure | r(211) | p | 95% Confidence interval | |
|---|---|---|---|---|
| | | | Lower | Upper |
| Neurite dispersion (ODI) | –0.09 | 0.17 | –0.23 | 0.05 |
| Neurite density | 0.12 | 0.09 | –0.01 | 0.24 |
| Fractional anisotropy | 0.13 | 0.05 | 0.01 | 0.26 |
| Mean diffusivity | –0.02 | 0.80 | –0.14 | 0.11 |
| Mean kurtosis | 0.11 | 0.12 | –0.03 | 0.24 |
| Diffusivities parallel | 0.12 | 0.09 | –0.03 | 0.25 |
| Diffusivities perpendicular | –0.08 | 0.23 | –0.20 | 0.03 |

Note. MT sat = Magnetisation Transfer saturation; ODI = Orientation Dispersion Index.

## Investigation of the standard DTI parameters extracted from the parahippocampal cingulum bundle

As reported in the main text, significant correlations between a number of microstructural measures from the parahippocampal cingulum bundle and autobiographical memory recall ability were evident. This was also the case for several of the standard DTI parameters – see *Appendix 1—table 19*, *Appendix 1—table 20* and *Appendix 1—figure 9*, *Appendix 1—figure 10* below.

**Appendix 1—table 19.** Means and standard deviations for the standard DTI parameters extracted from the parahippocampal cingulum bundle.

| Measure | Mean | Standard deviation |
|---|---|---|
| Fractional anisotropy | 0.466 | 0.053 |
| Mean diffusivity ($10^{-3}$ mm$^2$/s) | 0.931 | 0.041 |
| Mean kurtosis | 0.779 | 0.122 |
| Diffusivities parallel ($10^{-3}$ mm$^2$/s) | 1.479 | 0.071 |
| Diffusivities perpendicular ($10^{-3}$ mm$^2$/s) | 0.656 | 0.057 |

**Appendix 1—table 20.** Partial correlations between the standard DTI parameters extracted from the parahippocampal cingulum bundle and autobiographical memory recall ability (internal details).

| Measure | r(211) | p | 95% Confidence interval | |
|---|---|---|---|---|
| | | | Lower | Upper |
| Fractional anisotropy | 0.20 | 0.003* | 0.07 | 0.32 |
| Mean diffusivity | –0.02 | 0.72 | –0.15 | 0.11 |
| Mean kurtosis | 0.08 | 0.23 | –0.05 | 0.21 |
| Diffusivities parallel | 0.19 | 0.005* | 0.06 | 0.32 |

*Appendix 1—table 20 Continued on next page*

*Appendix 1—table 20 Continued*

| | | | 95% Confidence interval | |
| Measure | r(211) | p | Lower | Upper |
| --- | --- | --- | --- | --- |
| Diffusivities perpendicular | 0.15 | 0.03 | –0.27 | –0.02 |

* p < 0.017 (two-sided Bonferroni corrected threshold).

Specifically, partial correlation analyses, with age, gender, scanner and the number of voxels in the ROI included as covariates revealed a significant positive correlation between internal details and fractional anisotropy (FA) (*Appendix 1—figure 9A*; r(211) = 0.20, p=0.003, 95% CI=0.07, 0.32). This relationship was specific to internal details, and was not evident for the external details control measure (*Appendix 1—figure 9B*; r(211) = –0.06, p=0.39, 95% CI=–0.19, 0.07). Direct comparison of the two correlations confirmed a significant difference between them, showing that parahippocampal cingulum bundle FA was related to internal details to a greater extent than external details (*Appendix 1—figure 9C*; mean r difference = 0.26 (95% CI=0.10, 0.44), z=3.08, p=0.002).

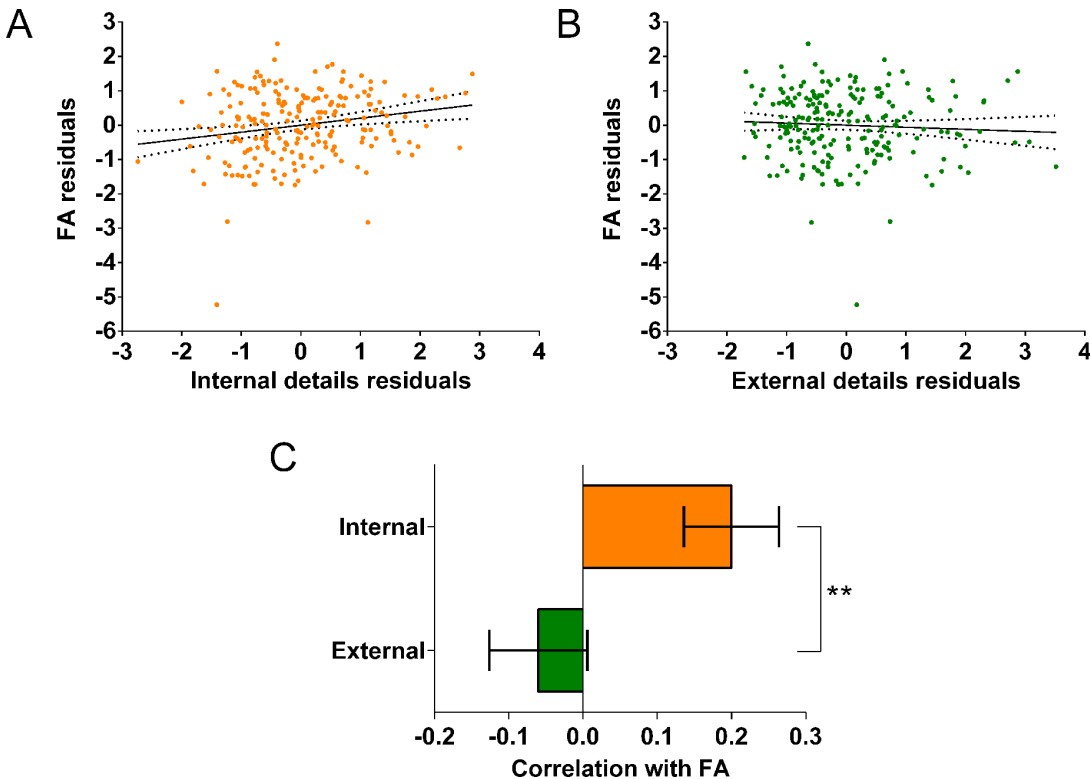

**Appendix 1—figure 9.** Fractional anisotropy (FA) and the parahippocampal cingulum bundle. The relationships between parahippocampal cingulum bundle FA and autobiographical memory recall ability (internal details), and the control measure (external details) are shown. (**A**) There was a significant positive correlation between FA and internal details (dashed lines indicate the confidence intervals). (**B**) There was no significant relationship between FA and external details. (**C**) Bar chart showing the partial correlation coefficients (with standard errors) between FA and internal and external details. There was a significant difference between the correlations when they were directly compared; **p<0.01. Data points for this figure are provided in *Supplementary file 2*.

In addition, a significant positive correlation between diffusivities parallel and the number of internal details was also apparent (*Appendix 1—figure 10A*; r(211) = 0.19, p=0.005, 95% CI=0.06, 0.32). As with FA, no significant relationship was observed between diffusivities parallel and external details (*Appendix 1—figure 10B*; r(211) = –0.048, p=0.49, 95% CI=–0.19, 0.10). Direct comparison of the correlations confirmed that diffusivities parallel was related to internal details to a greater extent than external details (*Appendix 1—figure 10C*; mean r difference = 0.24 (95% CI=0.07, 0.41), z=2.81, p=0.0049).

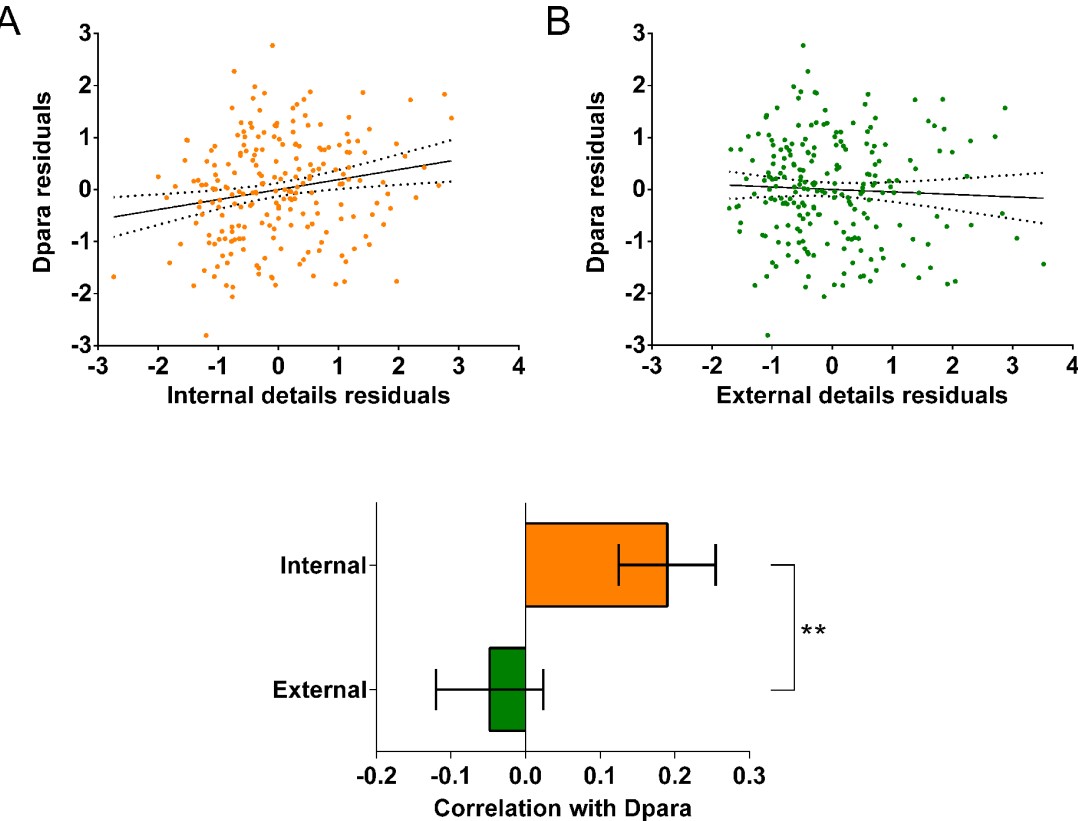

**Appendix 1—figure 10.** Dpara and the parahippocampal cingulum bundle. The relationships between parahippocampal cingulum bundle diffusivities parallel (Dpara) and autobiographical memory recall ability (internal details), and the control measure (external details) are shown. (**A**) There was a significant positive correlation between Dpara and internal details (dashed lines indicate the confidence intervals). (**B**) There was no significant relationship between Dpara and external details. (**C**) Bar chart showing the partial correlation coefficients (with standard errors) between Dpara and internal and external details. There was a significant difference between the correlations when they were directly compared; **p<0.01. Data points for this figure are provided in *Supplementary file 2*.

In contrast, no relationships were observed when examining the partial correlations between either internal or external details and mean diffusivity (internal: $r(211) = -0.02$, p=0.72, 95% CI=−0.15, 0.11; external: $r(211) = 0.01$, p=0.86, 95% CI=−0.11, 0.14), the mean kurtosis (internal: $r(211) = 0.08$, p=0.23, 95% CI=−0.05, 0.21; external: $r(211) = 0.03$, p=0.62, 95% CI=−0.01, 0.16) or diffusivities perpendicular (internal: $r(211) = -0.15$, p=0.03, 95% CI=−0.27,−0.02; external: $r(211) = 0.44$, p=0.52, 95% CI=−0.08, 0.17), when using the corrected (p<0.017) threshold. This suggests that none of these parameters were strongly associated with individual differences in autobiographical memory recall.

## Investigation of the parahippocampal cingulum bundle MR g-ratio and magnetisation transfer saturation values and the laboratory-based memory tests

Investigation of associations between the MR g-ratio and magnetisation transfer saturation values and the eight laboratory-based memory tests was also undertaken. No relationships were evident for any test, even when using an uncorrected p<0.05 threshold, see *Appendix 1—table 21*.

**Appendix 1—table 21.** Partial correlations between the parahippocampal cingulum bundle MR g-ratio and magnetisation transfer saturation values and the laboratory-based memory tests.

| Measure | r(211) | p | 95% Confidence interval | |
|---|---|---|---|---|
| | | | Lower | Upper |
| **MR g-ratio** | | | | |
| Logical Memory immediate recall | 0.01 | 0.93 | –0.13 | 0.13 |
| Logical Memory delayed recall | –0.05 | 0.46 | –0.18 | 0.08 |
| RAVLT immediate recall | –0.06 | 0.41 | –0.18 | 0.07 |
| RAVLT delayed recall | –0.09 | 0.17 | –0.21 | 0.04 |
| Rey-Osterrieth Complex Figure delayed recall | 0.07 | 0.34 | –0.06 | 0.19 |
| Warrington RMT for Words | 0.02 | 0.80 | –0.11 | 0.16 |
| Warrington RMT for Faces | –0.04 | 0.53 | –0.17 | 0.10 |
| Dead or Alive Test | 0.02 | 0.79 | –0.11 | 0.15 |
| **Magnetisation transfer saturation** | | | | |
| Logical Memory immediate recall | 0.07 | 0.35 | –0.06 | 0.19 |
| Logical Memory delayed recall | 0.01 | 0.91 | –0.12 | 0.14 |
| RAVLT immediate recall | –0.01 | 0.90 | –0.14 | 0.14 |
| RAVLT delayed recall | –0.02 | 0.80 | –0.14 | 0.11 |
| Rey-Osterrieth Complex Figure delayed recall | –0.09 | 0.21 | –0.23 | 0.05 |
| Warrington RMT for Words | –0.09 | 0.21 | –0.22 | 0.04 |
| Warrington RMT for Faces | –0.05 | 0.49 | –0.17 | 0.09 |
| Dead or Alive Test | 0.11 | 0.11 | –0.01 | 0.23 |

Note. RAVLT = Rey Auditory Verbal Learning Test; RMT = Recognition Memory Test.

