## [Editor Report]

In this paper, the authors show that autobiographical memory recall is related to a specific biophysical property of the parahippocampal cingulum bundle, the MR g-ratio. The paper presents compelling data supporting a fundamental new insight about the relationship between autobiographical memory and its underlying neural anatomy. The paper will be of strong interest to memory researchers as well as neuroscientists studying associations between brain structure and cognitive processes more generally.

---

## [Decision Letter]

**Decision letter after peer review:**

Thank you for submitting your article "Signalling speed along a key white matter tract influences memory recall ability in humans" for consideration by *eLife*. Your article has been reviewed by 3 peer reviewers, and the evaluation has been overseen by a Reviewing Editor and Laura Colgin as the Senior Editor. The reviewers have opted to remain anonymous.

We are sorry to say that, after consultation with the reviewers, we have decided that the work as described in the current manuscript will not be considered for publication by *eLife*. However, *eLife* might consider a very substantial revision, if the authors are interested in undertaking it. The new version would need to address all of the reviewers' concerns, including incorporating reviewer 1's suggestion to report correlations between white matter tracts and all cognitive measures collected, addressing concerns about the g-ratio measurement, as well as changing claims about measuring signaling speed and removing language suggesting causal relationships.

*Reviewer #1 (Recommendations for the authors):*

My main recommendation is that the authors report the raw correlations between other memory tests and the MR g-ratios of their candidate white matter tracts (assuming that this is possible). This will either show that increased g-ratios in the parahippocampal cingulum bundle are a general feature of individuals who perform better on memory tests, or it is something more specific to the scores on the Autobiographical Memory Interview. It might be the case that other tracts show associations with other tests. Any of these scenarios would help contextualize the results of the paper as they stand.

I am not particularly concerned about correcting for multiple comparisons or arbitrary thresholds for significance. I think that the study is important, novel, and well carried out and that the findings will be important to researchers in several fields. If it is the case that reliable associations between cognitive measures and the MR g-ratio cannot be established in a sample of 217 people then this is really useful information!

I have probably misunderstood the interpretation of the findings with respect to the examples shown in figure 1. It seemed that the authors interpreted their positive correlation between internal details and the g-ratio in the parahippocampal cingulum bundle as evidence for option D – people with better memory recall scores are likely to have larger inner axon diameters and relatively smaller increases in myelin. However, since example 1E involves no change in the g-ratio, I couldn't understand why the finding of no relationship between memory recall ability and g-ratio from the fornix or uncinate fasciculus was not taken as support for option E. It might help if the authors could clarify which situation(s) would lead to them accepting option E as the best explanation of their results.

The following paper by Hodgetts et al., seems very relevant – the present study fails to replicate their findings related to the fornix.

Hodgetts, C. J., Postans, M., Warne, N., Varnava, A., Lawrence, A. D., and Graham, K. S. (2017). Distinct contributions of the fornix and inferior longitudinal fasciculus to episodic and semantic autobiographical memory. cortex, 94, 1-14.

*Reviewer #2 (Recommendations for the authors):*

– Following what I mentioned in the public review, given the lack of any other measure to quantitatively or even qualitatively assess either conduction velocity/delays or axon size, the authors should refrain from directly claiming that larger axons mean better memory recall abilities, which instead is done in the abstract (page 2, "(…) two tract features seemed to favour better memory retrieval – large inner axon diameters (…)" – despite the axon diameters not being measured or estimated), and in the discussion (page 18, "(…) we were able to identify two particular features (…) that might favour better memory recall, namely larger inner axon diameters (…)").

– Related to the previous point, a suggestion that I have is to include, among the exhaustive comparisons performed in the appendix, the ones with the MTsat values. As an absolute measure of myelin, MTsat could further help the authors by providing a complete picture of their results.

– It is rather peculiar that the abstract does not mention at all myelin (which in the end is the main protagonist of the study) and that does not briefly define the g-ratio. I understand the authors' focus on the conduction velocity, but these additions would surely benefit a more general audience.

– Related to the previous point, the authors may also consider rethinking the use of "signalling speed", which basically appears only in the title and in the abstract, in favour of "conduction velocity" which is used extensively in the text.

– On page 3, the authors stated: "The scope for signals to travel quickly and efficiently between brain regions could make all the differences for an individual's survival". Although the idea that faster conduction velocity constitutes an evolutionary advantage could be appealing and the peripheral nervous system (PNS) supports the concept extremely well (a very famous example is depicted in Wedel, Acta Palaeontologica 2011), there are several reasons to be cautious with such a statement. In contrast to what happens in the PNS (where faster propagation reflects in faster sensory processing and faster motor responses), in the central nervous system conduction phenomena serve the purpose of guaranteeing synchrony in the activity of different neuronal populations (Seidtl, Neuroscience 2014; Gollo et al., NeuroImage 2017; Pariz et al., Plos Computational Biology 2021) and the role of delays could be more subtle. Unless the authors can integrate the statement with some references to experimental evidence, I would remove such a statement.

– Still on page 3, the authors mentioned that the conduction velocity "is dependent upon two biological features – the axonal diameter and the presence and thickness of a myelin sheath". This picture is too approximative as experimental work (Arancibia-Carcamo et al., *eLife* 2017) and simulations (Drakesmith et al., NeuroImage 2019) have shown how several parameters influence conduction velocity, potentially even with a larger impact than the g-ratio. It would be appropriate for a more general audience to add more details. Similarly, on page 4 it would help to refer to saltatory propagation when explaining the effect of myelin as an insulator.

– On page 7, there is a bit of a leap between the section on the uses and potentials of the MRI-based g-ratio and the proposed application in autobiographical memory. It would help to provide a couple of sentences to link those together or to rearrange a bit the paragraph in-between pages 7 and 8, where at the very end the case for g-ratio and memory is made.

– The considerations in figure 1 made on pages 4-5 are at the moment very hard without referring to an actual relationship between conduction velocity, axonal diameter, and g-ratio. Even though the authors are not actually computing conduction velocity, it would help to refer to established formulas such as the one from Rushton (Rushton, Journal of Physiology 1951) or the one form Waxman and Bennet (Waxman and Bennet, Nature New Biology 1972).

– I noticed that in Figures 4 and 5 the scatterplots are based on the residuals rather than the actual g-ratio and neurite dispersion values. Can I ask what is the reason behind this choice? This representation makes actually harder to get a visual sense of the dynamics of the reported relationships.

– On page 18, the authors mentioned how these results can provide a new perspective on the underlying neural substrate of autographical memory, and stated: "This is especially welcome given the dearth of consistent findings linking hippocampal grey matter volume or microstructure with autobiographical memory recall ability in the healthy population". Can the authors expand their argument here? I am missing the link between the current findings (on white matter microstructure) with previous ones on grey matter features.

– On page 22, there is a reference to the CONNECTOM MRI scanners (misspelled as "connectome MRI scanners") – as the general audience is not familiar with those it would help to add a brief sentence explaining what they are and maybe a reference (for example, Jones et al., NeuroImage 2018).

– Can the authors comment on the advantages of the proposed ROI-based approach compared with a tract-based approach?

– I may have missed it, but I am not sure of what is the reason for the fornix tract being "not probabilistic" (page 29) – are not all the tract ROIs obtained from the John Hopkins probabilistic atlas?

– I could not find any mention of the main methodological limitations (e.g., the use of signal fractions rather than volume fractions, the bias towards larger axons, and the need for single-point calibration). There is only a brief mention in the appendix on page 43. I think these limitations should be part of the discussion to put these results in the right context.

*Reviewer #3 (Recommendations for the authors):*

I have some suggestions the authors may consider, presented in chronological order.

Title

The title suggests a causal relationship, while this is based on an association. Maybe better to use a neutral word, i.e. replace "influences" with e.g. "is associated with" or something similar?

As I understand MR g-ratio, it reflects structural aspects of nerve fibers, and although probably related to conduction velocity/ signaling speed, it is not a direct measure of this. Hence, I believe the title is overstating this relationship by referring to "signalling speed" directly. I fully understand the need for short and focused titles, but I believe this is a bit misleading. The same point applies to a couple of other places in the manuscript too. Of course fine to discuss this as an interpretation.

Abstract

Not all readers are probably familiar with the MR g-ratio, so maybe it can be defined in the abstract?

Since this is a cross-sectional study, I suggest avoiding phrases that imply change, e.g. "increased MR g-ratio". Maybe replace it with something neutral, such as "higher MR g-ratio"? The same is the case in several places in the manuscript, e.g. ln 195.

As with the title, the results reflect associations, not causation. This does not render them less interesting, but I would recommend replacing "favour better memory …" with e.g. "is associated with better memory …". I also wonder whether "offer a new perspective on drivers of individual differences in memory recall ability" is overstating a bit, since (a) we don't know whether g-ratio (or signal speed) is driving anything, and (b) this is a rather well-established theory (as also explained in the introduction, and it is stated ln 44 "it is perhaps no surprise that speed may be of the essence")?

Introduction

ln 51-54: Why was aging discussed? Reading the introductory paragraph, I expected aging to be part of the study, which it was not.

Ln 123-125: "While some healthy individuals can recollect decades-old autobiographical memories with great richness and clarity, others struggle to recall what they did last weekend" Yes, but as the authors surely know, this reflects different memory systems or processes, and therefore can happen within the same individual too, not only as a feature of inter-individual differences.

Ln 195-200: Figure 1 clearly illustrates different scenarios, which is very useful. However, the hypothesis that better autobiographical memory is related to more myelin could also be consistent with no relationship with the MR g-ratio if the axon diameter is larger in participants with a good memory, right? And vice versa for the hypothesis about axon diameter? I don't see how the ratio measure can distinguish between the different accounts since the measures per definition are related to two possibly independent properties which both can be of importance? The authors multiple times mention that other relevant metrics, such as those derived from DTI, lack biological specificity. But the MR g-ratio measure seems also to be unspecific in this regard. If axon diameter and myelin thickness can be estimated separately, why not use them directly in the analyses instead of the MR g-ratio? There may be a methodological reason for this, but I think this could be more clearly explained as MR g-ratio is not often reported in the literature and so will be novel to many readers. Relatedly, although there are methods papers on this topic, maybe a brief justification of how can ratios between these small compartments on the micrometer scale can be calculated from voxels on the millimeter scale already in the introduction?

Results

I understand that the authors don't want to provide the full description of results for FA and MD in the main text, but it would be convenient to report a brief summary of the main results also in the text. As far as I can see from Appendix 1, none even approach significance for fornix and uncinated fasciculus, so maybe at least state that?

This may be a matter of taste, but I would suggest moving the section on why the parahippocampal cingulum bundle should be important for autobiographical memory from the Results section (ln 236 and onwards) to the introduction or the discussion.

Ln 279: A positive relationship between MR g-ratio and memory is interpreted as reflecting larger inner axon diameters. Ref Figure 1, this can also be due to thinner myelin sheets, right, as it is a ratio measure? Multiple comparison correction was mentioned for some analyses above, was this done here two, and if so, how? Would be useful to state explicitly.

Inspecting the Appendix, it seems that the relationships between autobiographic memory and PCB FA and MD were also significant, with r-values that were at least as large as for MR g-ratio. Why not include this in the main text? These are metrics of interest to many researchers. You could also run a multiple regression analysis, and test whether each metric gives unique information or whether they are redundant. I think this would be a very useful addition to the manuscript, regardless of the outcome. If MR g-ratio is still significantly related to autobiographical memory, this would strengthen the main argument in the paper. If not, we will know more about the interrelationships between these different tract metrics, which also will be very useful.

Discussion

Ln 336: Is it correct to say that the MR g-ratio – memory correlation represents "neural instantiation of autobiographical memory"? To me, it seems to overstate the results, as this is an association, and we do not know whether there is any causality involved. The same goes for the last part of the sentence "drivers of individual differences in recall ability". I think the results are of sufficient interest, without such statements.

Methods, Participants

Ln 456-457: "People with hobbies or vocations known to be associated with the hippocampus (e.g. licenced London taxi drivers) were excluded." Could you please be a little more specific, most activities will be associated with the hippocampus? How many were excluded based on this?

[Editors’ note: further revisions were suggested prior to acceptance, as described below.]

The authors put an amazing effort into taking into account the reviewers' comments and into improving the paper with further details and analysis. The finding that magnetization transfer saturation does not correlate with the score of interest while the g-ratio instead does is quite interesting - I think that for the sake of fully characterizing the relationship what is missing at the moment is correlating the isotropic volume fraction with the recall score: since the g-ratio is computed using MTsat, Viso and Vic, double-checking that a correlation does not exist with Viso is the last step needed to show that indeed is the combination of all these measures that is meaningful rather than the single measures themselves.

---

## [Author Response]

Reviewer #1 (Recommendations for the authors):My main recommendation is that the authors report the raw correlations between other memory tests and the MR g-ratios of their candidate white matter tracts (assuming that this is possible). This will either show that increased g-ratios in the parahippocampal cingulum bundle are a general feature of individuals who perform better on memory tests, or it is something more specific to the scores on the Autobiographical Memory Interview. It might be the case that other tracts show associations with other tests. Any of these scenarios would help contextualize the results of the paper as they stand.

As noted in our response to the Reviewer’s Public Review, our main interest over many years, and hence the focus of this study, is autobiographical memory recall because it directly relates to how people function in real life. We used the current, gold standard approach to measuring autobiographical memory recall ability (Levine’s Autobiographical Interview). Nevertheless, at the Reviewer’s request we have now analysed data from eight additional laboratory-based memory tests. Scores from these tests did not correlate with the MR g-ratio for any of our three white matter tracts of interest. Recall of vivid, detailed, multimodal, autobiographical memories may rely on inter-regional connectivity to a greater degree than simpler, more constrained laboratory-based memory tests.

Therefore, as well as speaking to conduction velocity, our findings now also contribute to wider discussions about real-world compared to laboratory-based memory tests. We thank the Reviewer again for making the excellent suggestion to include these additional data, analyses and discussion points, as noted below.

Introduction, p. 10: “All participants underwent the widely-used Autobiographical Interview (Levine et al., 2002). This task, which is the gold standard in the field, provides a detailed metric characterising a person’s ability to recall real-life past experiences, as well as providing a useful control measure. We also examined performance on another eight standard laboratory-based memory tests. Their inclusion allowed us to ascertain whether any relationships with conduction velocity were specific to recollecting detailed autobiographical memories from real life, or were also applicable to the recall of more constrained laboratory-based stimuli.”

Introduction, pp. 11-12: “Finally, given that a previous functional MRI meta-analysis identified different neural substrates associated with autobiographical memory recall and the recall of laboratory-based stimuli (McDermott et al., 2009; see also Maguire, 2001, 2012; Miller et al., 2022; Mobbs et al., 2021; Nastase et al., 2020; Spiers and Maguire, 2007), we expected that associations with the MR g-ratio might be specific to autobiographical memory recall. We reasoned that detailed, multimodal, autobiographical memories may rely on inter-regional connectivity to a greater degree than simpler, more constrained laboratory-based memory tests.”

Results, pp. 12-13:

“Laboratory-based memory test performance

While our main interest was in autobiographical memory recall, eight commonly used laboratory-based memory tasks were also administered. Their inclusion allowed us to establish whether any associations identified with the main microstructure measure, MR g-ratio, were specific or not to the recollection of real-life autobiographical memories.

The ability to recall a short narrative was examined using the immediate and delayed recall tests of the Logical Memory subtest of the Wechsler Memory Scale IV (Wechsler, 2009). Across participants, the mean immediate recall scaled score was 12.95 (SD = 2.09, range = 6-18) and the mean delayed recall scaled score was 12.58 (SD = 2.62, range = 6-19). Verbal list recall ability was assessed using the immediate and delayed recall of the Rey Auditory Verbal Learning Test (see Strauss et al., 2006). The mean immediate recall (aggregate) score was 58.82 (SD = 7.42, range = 33-73) and the mean delayed recall score was 12.92 (SD = 2.17, range = 6-15). Visuospatial recall ability was examined using the delayed recall of the Rey–Osterrieth Complex Figure (Rey, 1941), with a mean delayed recall score of 22.28 (SD = 5.71, range = 8.5-35). Recognition memory ability was tested using the Warrington Recognition Memory Tests for Words and Faces (Warrington, 1984). The mean recognition memory scaled score for words was 12.75 (SD = 2.06, range = 3-15) and for faces was 11.00 (SD = 3.33, range = 3-18). Finally, participants also completed the “Dead or Alive” task which probes general knowledge about whether famous people have died or are still alive, providing a measure of semantic memory (Kapur et al., 1989). The mean accuracy performance on this test was 81.32% (SD = 8.44, range = 57.14%-97.26%).”

Results, p. 14: “There were no significant associations between fornix or uncinate fasciculus MR g-ratios or magnetisation transfer saturation values and any of the laboratory-based memory tests (see Appendix 1– tables 5 and 6 for full details, source data are available in Supplementary file 1).”

Results, p. 18:

“No relationship between microstructure measures of the parahippocampal cingulum bundle and scores on laboratory-based memory tests

Finally, we tested for associations between the key microstructure measures and performance on the eight laboratory-based memory tasks. No relationships with the MR g-ratio or magnetisation transfer saturation values were evident for any task, even when using an uncorrected p < 0.05 threshold (Appendix 1-table 21, source data are available in Supplementary file 2). This suggests that the parahippocampal cingulum bundle MR g-ratio, and by inference conduction velocity, was specifically associated with recall of autobiographical memories from real life rather than performance on the more constrained laboratory-based memory tests.”

Discussion, p. 24: “Moreover, no relationships were identified with any of the eight laboratory-based memory tasks examined, highlighting the specific nature of the relationship between parahippocampal cingulum bundle MR g-ratio and, by inference, conduction velocity, and autobiographical memory recall ability. It may be that vivid, detailed, multimodal, autobiographical memories rely on inter-regional connectivity, particularly that supported by the parahippocampal cingulum bundle, to a greater degree than simpler, more constrained laboratory-based memory tests. This result aligns with previous work involving an fMRI meta-analysis which showed that recall of autobiographical memories and laboratory-based memory stimuli were associated with substantially different neural substrates (McDermott et al., 2009). More generally, our findings add to the increasingly-recognised importance of studying real-world cognition in order to fully characterise brain-behaviour relationships (Maguire, 2001, 2012; Miller et al., 2022; Mobbs et al., 2021; Nastase et al., 2020; Spiers and Maguire, 2007).”

Materials and methods, pp. 28-29:

“Laboratory-based memory tests

Eight laboratory-based memory tasks were also administered to participants. These are standard memory tests that are often used in neuropsychological settings. Tasks were performed and scored in line with their standardised and published protocols.

The ability to recall a short narrative was examined using the immediate and delayed recall tests of the Logical Memory subtest of the Wechsler Memory Scale IV (Wechsler, 2009). Verbal list recall was assessed using the immediate and delayed recall tests of the Rey Auditory Verbal Learning Test (see Strauss et al., 2006). Visuospatial recall was examined using the delayed recall of the Rey–Osterrieth Complex Figure (Rey, 1941). Recognition memory was investigated using the Warrington Recognition Memory Tests for Words and Faces (Warrington, 1984). Finally, participants also underwent the “Dead or Alive” task which probes general knowledge about whether famous individuals have died or are still alive, providing a measure of semantic memory (Kapur et al., 1989).”

Materials and methods, p. 35: “We also investigated the relationship between the MR g-ratio and the magnetisation transfer saturation values and eight laboratory-based memory tests. As with the main analyses, partial correlations were performed between the MR g-ratio and magnetisation transfer saturation values and the outcome measures of the memory tests with age, gender, scanner and the number of voxels in a tract ROI included as covariates, and bootstrapping performed 10,000 times to calculate confidence intervals.”

I am not particularly concerned about correcting for multiple comparisons or arbitrary thresholds for significance. I think that the study is important, novel, and well carried out and that the findings will be important to researchers in several fields. If it is the case that reliable associations between cognitive measures and the MR g-ratio cannot be established in a sample of 217 people then this is really useful information!

Thank you.

I have probably misunderstood the interpretation of the findings with respect to the examples shown in figure 1. It seemed that the authors interpreted their positive correlation between internal details and the g-ratio in the parahippocampal cingulum bundle as evidence for option D – people with better memory recall scores are likely to have larger inner axon diameters and relatively smaller increases in myelin. However, since example 1E involves no change in the g-ratio, I couldn't understand why the finding of no relationship between memory recall ability and g-ratio from the fornix or uncinate fasciculus was not taken as support for option E. It might help if the authors could clarify which situation(s) would lead to them accepting option E as the best explanation of their results.

We are grateful for the opportunity to clarify the situation regarding option E (Figure 2A black arrow, and Figure 2E), and agree it requires further unpacking.

In fact, there are three main scenarios. (1) A decrease in MR g-ratio values. This would suggest that faster conduction velocity is due to greater thickness of the myelin sheath, with the inner axon diameter remaining constant (Figure 2A blue arrow, and Figure 2B). (2) A decrease in MR g-ratio values, but to a lesser extent than that observed in the first scenario. This would suggest that faster conduction velocity is due primarily to greater myelin sheath thickness, but one that is also accompanied by a larger inner axon diameter (Figure 2A red arrow, and Figure 2C). (3) An increase in MR g-ratio values. This would suggest that faster conduction velocity is predominantly due to a larger inner axon diameter, with only small increases in myelin thickness also being present (Figure 2A orange arrow, and Figure 2D).

A fourth scenario also exists in which constant MR g-ratio values could be associated with faster conduction velocity. This would occur when both the inner axon diameter and myelin thickness change proportionally to each other (Figure 2A black arrow, and Figure 2E). However, this scenario could also mean that there is no variation in conduction velocity. One way to increase interpretability in this situation, is by examining scans optimised to assess myelination (such as magnetisation transfer saturation maps). Observing no relationship with the MR g-ratio, but a relationship with magnetisation transfer saturation values, would suggest a proportional change in the underlying myelin and, consequently, variation in conduction velocity. By contrast, no relationship with either the MR g-ratio or magnetisation transfer saturation values would suggest that there was no change in the underlying microstructure, and therefore no variation in conduction velocity. In fact, we observed no associations between the magnetisation transfer saturation values and autobiographical memory recall, suggesting no relationships with myelin or conduction velocity.

We now explicitly consider Figure 2E (black arrow) and its interpretation in the revised manuscript, as detailed below.

Introduction, p. 4: “The conduction velocity of an axon is, therefore, not determined only by the axon diameter, but also by the relationship between the axon diameter and the thickness of the surrounding myelin sheath, a measure known as the g-ratio (Chomiak and Hu, 2009; Rushton, 1951; Schmidt and Knösche, 2019). Specifically, Rushton (1951) derived an equation: conduction velocity∝d−ln(g), where d is the inner axon diameter, g is the g-ratio = d/D, and D is the outer fibre (axon plus myelin sheath) diameter. In other words, the g-ratio is computed as the ratio of the inner axon diameter relative to that of the axon plus the myelin sheath that encases it (Figure 1).”

Introduction, p. 5: “Of note, the MR g-ratio has been found to associate well with estimates of axonal conduction velocity (Berman et al., 2019; Drakesmith et al., 2019). Consequently, the MR g-ratio provides a non-invasive MRI method that can associate in vivo structural neuroimaging of humans with axonal conduction velocity.”

Introduction, p. 7: “We note, for completeness, that a fourth scenario also exists where constant MR g-ratio values could be associated with faster conduction velocity. This would occur when both the inner axon diameter and myelin thickness change proportionally to each other (Figure 2A black arrow, and Figure 2E). However, this scenario could also mean that there is no variation in conduction velocity. One way to increase interpretability in this situation, is by examining scans optimised to assess myelination (such as magnetisation transfer saturation values). Observing no relationship with the MR g-ratio, but a relationship with magnetisation transfer saturation values, would suggest a proportional change in the underlying myelin and, consequently, variation in conduction velocity. By contrast, no relationship with either the MR g-ratio or magnetisation transfer saturation values would suggest that there was no change in the underlying microstructure and therefore no variation in conduction velocity.”

Introduction, p. 9: “The relationship of MR g-ratio to the underlying microstructure as outlined in Figure 2, would further guide us as to whether any significant effects were more likely to be associated with the extent of myelination or the size of the inner axon diameter of the fibres in these three white matter tracts. These analyses were augmented by examining whether magnetisation transfer saturation values of the pathways (assessing myelination) were associated with autobiographical memory recall ability.”

Introduction, pp. 10-11: “Focusing on the fornix, uncinate fasciculus and parahippocampal cingulum bundle, we predicted that variations in the MR g-ratio from some or all of these tracts would be associated with autobiographical memory recall ability. Such a finding would, for the first time, suggest a link between variations in white matter tract conduction velocity and individual differences in autobiographical memory recall.

Under the assumption that a significant relationship between the MR g-ratio and autobiographical memory recall represents an association with faster conduction velocity (e.g. Brancucci, 2012; Dicke and Roth, 2016; Miller, 1994; Reed and Jensen, 1992), and aided by the analysis of the magnetisation transfer saturation values, we further sought to evaluate the scenarios presented in Figure 2. Specifically, we asked whether autobiographical memory recall ability was more likely to be associated with inner axon diameters or myelin thickness. A negative relationship between the MR g-ratio and autobiographical memory recall ability (Figure 2A blue and red arrows, and Figures 2B and 2C), along with a positive relationship between magnetisation transfer saturation and autobiographical memory recall ability, would suggest that myelin thickness was more relevant. By contrast, a positive relationship between the MR g-ratio and autobiographical memory recall ability (Figure 2A orange arrow, and Figure 2D) would highlight the potential relevance of inner axon diameters. Observing no associations between the MR g-ratio or magnetisation transfer saturation values and autobiographical memory recall, would suggest no relationships with the underlying microstructure or conduction velocity, and would speak against the possibility of a proportional change in microstructure (Figure 2A black arrow, and Figure 2E).”

Results, p. 16: “To further aid the interpretation of the relationship with MR g-ratio in terms of the underlying axonal microstructure, we also investigated whether the magnetisation transfer saturation values (assessing myelination) of the parahippocampal cingulum bundle were associated with autobiographical memory recall ability. We performed partial correlation analyses with the same covariates as before. Magnetisation transfer saturation values were not significantly related to either internal (r(211) = 0.00, p = 1.0, 95% CI = -0.12, 0.12), or external (r(211) = 0.14, p = 0.048, 95% CI = -0.06, 0.30) details.”

Discussion, p. 20: “The positive relationship we observed between the parahippocampal cingulum bundle MR g-ratio and autobiographical memory recall ability suggests that this effect was associated predominantly with larger inner axon diameter (Figure 2A orange arrow, and Figure 2D). By contrast, had a negative correlation between the MR g-ratio and autobiographical memory recall ability been identified, we could instead have inferred that myelination was the relevant microstructural feature. The lack of relationship between parahippocampal cingulum bundle magnetisation transfer saturation values (optimised to assess myelination) and autobiographical memory recall ability provided further corroboration that myelin may not have been influential in this context.”

Discussion, pp. 22-23: “We also examined two such tracts, the fornix and the uncinate fasciculus, but in both cases no significant relationships between any of our neuroimaging metrics and autobiographical memory recall ability were evident in our large cohort of young healthy adults. As outlined in Figure 2, there was one possible scenario where a null relationship with the MR g-ratio could reflect an association with conduction velocity – if both the inner axon diameter and myelin thickness increased proportionally to each other (Figure 2A black arrow, and Figure 2E). However, we found no associations between autobiographical memory recall and both the MR g-ratio and magnetisation transfer saturation values of the fornix and uncinate fasciculus, which speaks against this explanation.”

The following paper by Hodgetts et al., seems very relevant – the present study fails to replicate their findings related to the fornix.Hodgetts, C. J., Postans, M., Warne, N., Varnava, A., Lawrence, A. D., and Graham, K. S. (2017). Distinct contributions of the fornix and inferior longitudinal fasciculus to episodic and semantic autobiographical memory. cortex, 94, 1-14.

Thank you for highlighting this paper, which we now reference.

Discussion, p. 23: “In terms of standard DTI parameters, unlike a previous study (Hodgetts et al., 2017), we did not find a relationship between fornix fractional anisotropy and autobiographical memory recall ability, which may be due to the larger sample that we examined.”

Reviewer #2 (Recommendations for the authors):– Following what I mentioned in the public review, given the lack of any other measure to quantitatively or even qualitatively assess either conduction velocity/delays or axon size, the authors should refrain from directly claiming that larger axons mean better memory recall abilities, which instead is done in the abstract (page 2, "(…) two tract features seemed to favour better memory retrieval – large inner axon diameters (…)" – despite the axon diameters not being measured or estimated), and in the discussion (page 18, "(…) we were able to identify two particular features (…) that might favour better memory recall, namely larger inner axon diameters (…)").

Changes have been made throughout the manuscript to avoid any direct claims of this nature, and to emphasise inferences and associations, where appropriate.

– Related to the previous point, a suggestion that I have is to include, among the exhaustive comparisons performed in the appendix, the ones with the MTsat values. As an absolute measure of myelin, MTsat could further help the authors by providing a complete picture of their results.

As noted in our response to the Reviewer’s Public Review, thank you, we think this is an excellent suggestion. We now include analyses examining relationships between the magnetisation transfer saturation values and autobiographical memory recall ability for all tracts. No significant associations were evident. We also note in the revised manuscript that these findings support the inference that there seems to be no relationship between conduction velocity and autobiographical memory recall for the fornix and uncinate fasciculus. It also aligns with our assumption that here, relationships with the MR g-ratio were indicative of faster, not slower, conduction velocity, and that for the parahippocampal cingulum bundle inner axon diameter might be of more relevance than myelin thickness in relation to autobiographical memory recall ability.

Introduction, p. 7: “We note, for completeness, that a fourth scenario also exists where constant MR g-ratio values could be associated with faster conduction velocity. This would occur when both the inner axon diameter and myelin thickness change proportionally to each other (Figure 2A black arrow, and Figure 2E). However, this scenario could also mean that there is no variation in conduction velocity. One way to increase interpretability in this situation, is by examining scans optimised to assess myelination (such as magnetisation transfer saturation values). Observing no relationship with the MR g-ratio, but a relationship with magnetisation transfer saturation values, would suggest a proportional change in the underlying myelin and, consequently, variation in conduction velocity. By contrast, no relationship with either the MR g-ratio or magnetisation transfer saturation values would suggest that there was no change in the underlying microstructure and therefore no variation in conduction velocity.”

Introduction, p. 9: “The relationship of MR g-ratio to the underlying microstructure as outlined in Figure 2, would further guide us as to whether any significant effects were more likely to be associated with the extent of myelination or the size of the inner axon diameter of the fibres in these three white matter tracts. These analyses were augmented by examining whether magnetisation transfer saturation values of the pathways (assessing myelination) were associated with autobiographical memory recall ability.”

Introduction, p. 11: “Under the assumption that a significant relationship between the MR g-ratio and autobiographical memory recall represents an association with faster conduction velocity (e.g. Brancucci, 2012; Dicke and Roth, 2016; Miller, 1994; Reed and Jensen, 1992), and aided by the analysis of the magnetisation transfer saturation values, we further sought to evaluate the scenarios presented in Figure 2. Specifically, we asked whether autobiographical memory recall ability was more likely to be associated with inner axon diameters or myelin thickness. A negative relationship between the MR g-ratio and autobiographical memory recall ability (Figure 2A blue and red arrows, and Figures 2B and 2C), along with a positive relationship between magnetisation transfer saturation and autobiographical memory recall ability, would suggest that myelin thickness was more relevant. By contrast, a positive relationship between the MR g-ratio and autobiographical memory recall ability (Figure 2A orange arrow, and Figure 2D) would highlight the potential relevance of inner axon diameters. Observing no associations between the MR g-ratio or magnetisation transfer saturation values and autobiographical memory recall, would suggest no relationships with the underlying microstructure or conduction velocity, and would speak against the possibility of a proportional change in microstructure (Figure 2A black arrow, and Figure 2E).”

Results, p. 14: “There were no significant associations between fornix or uncinate fasciculus MR g-ratios or magnetisation transfer saturation values and any of the laboratory-based memory tests (see Appendix 1– tables 5 and 6 for full details, source data are available in Supplementary file 1).”

Results, p. 16: “To further aid the interpretation of the relationship with MR g-ratio in terms of the underlying axonal microstructure, we also investigated whether the magnetisation transfer saturation values (assessing myelination) of the parahippocampal cingulum bundle were associated with autobiographical memory recall ability. We performed partial correlation analyses with the same covariates as before. Magnetisation transfer saturation values were not significantly related to either internal (r(211) = 0.00, p = 1.0, 95% CI = -0.12, 0.12), or external (r(211) = 0.14, p = 0.048, 95% CI = -0.06, 0.30) details.”

Results, p. 18:

“No relationship between microstructure measures of the parahippocampal cingulum bundle and scores on laboratory-based memory tests

Finally, we tested for associations between the key microstructure measures and performance on the eight laboratory-based memory tasks. No relationships with the MR g-ratio or magnetisation transfer saturation values were evident for any task, even when using an uncorrected p < 0.05 threshold (Appendix 1-table 21, source data are available in Supplementary file 2). This suggests that the parahippocampal cingulum bundle MR g-ratio, and by inference conduction velocity, was specifically associated with recall of autobiographical memories from real life rather than performance on the more constrained laboratory-based memory tests.”

Discussion, p. 20: “The positive relationship we observed between the parahippocampal cingulum bundle MR g-ratio and autobiographical memory recall ability suggests that this effect was associated predominantly with larger inner axon diameter (Figure 2A orange arrow, and Figure 2D). By contrast, had a negative correlation between the MR g-ratio and autobiographical memory recall ability been identified, we could instead have inferred that myelination was the relevant microstructural feature. The lack of relationship between parahippocampal cingulum bundle magnetisation transfer saturation values (optimised to assess myelination) and autobiographical memory recall ability provided further corroboration that myelin may not have been influential in this context.”

Discussion, pp. 22-23: “We also examined two such tracts, the fornix and the uncinate fasciculus, but in both cases no significant relationships between any of our neuroimaging metrics and autobiographical memory recall ability were evident in our large cohort of young healthy adults. As outlined in Figure 2, there was one possible scenario where a null relationship with the MR g-ratio could reflect an association with conduction velocity – if both the inner axon diameter and myelin thickness increased proportionally to each other (Figure 2A black arrow, and Figure 2E). However, we found no associations between autobiographical memory recall and both the MR g-ratio and magnetisation transfer saturation values of the fornix and uncinate fasciculus, which speaks against this explanation.”

– It is rather peculiar that the abstract does not mention at all myelin (which in the end is the main protagonist of the study) and that does not briefly define the g-ratio. I understand the authors' focus on the conduction velocity, but these additions would surely benefit a more general audience.– Related to the previous point, the authors may also consider rethinking the use of "signalling speed", which basically appears only in the title and in the abstract, in favour of "conduction velocity" which is used extensively in the text.

The abstract has now been updated to include reference to myelination and to define the g-ratio. The phrase “signalling speed” has also been replaced in the title with “conduction velocity”. Thank you for these suggestions.

– On page 3, the authors stated: "The scope for signals to travel quickly and efficiently between brain regions could make all the differences for an individual's survival". Although the idea that faster conduction velocity constitutes an evolutionary advantage could be appealing and the peripheral nervous system (PNS) supports the concept extremely well (a very famous example is depicted in Wedel, Acta Palaeontologica 2011), there are several reasons to be cautious with such a statement. In contrast to what happens in the PNS (where faster propagation reflects in faster sensory processing and faster motor responses), in the central nervous system conduction phenomena serve the purpose of guaranteeing synchrony in the activity of different neuronal populations (Seidtl, Neuroscience 2014; Gollo et al., NeuroImage 2017; Pariz et al., Plos Computational Biology 2021) and the role of delays could be more subtle. Unless the authors can integrate the statement with some references to experimental evidence, I would remove such a statement.

We agree, and have now removed this statement from the Introduction.

– Still on page 3, the authors mentioned that the conduction velocity "is dependent upon two biological features – the axonal diameter and the presence and thickness of a myelin sheath". This picture is too approximative as experimental work (Arancibia-Carcamo et al., eLife 2017) and simulations (Drakesmith et al., NeuroImage 2019) have shown how several parameters influence conduction velocity, potentially even with a larger impact than the g-ratio. It would be appropriate for a more general audience to add more details. Similarly, on page 4 it would help to refer to saltatory propagation when explaining the effect of myelin as an insulator.

In the revised manuscript we now include additional details about how the conduction velocity of an axon is dependent upon its physiology, and what factors are measurable in vivo in humans. We also refer to saltatory propagation when explaining the effect of myelin as an insulator. We agree that these additions provide a more informative context.

Introduction, pp. 3-4:

“The conduction velocity of an axon is dependent upon the axon diameter, the presence and thickness of a myelin sheath, the distance between the nodes of Ranvier (periodic gaps in the myelin that facilitate action potential propagation), inter-nodal spacing, and electrical properties of the axonal and myelin membranes (Arancibia-Cárcamo et al., 2017; Drakesmith et al., 2019; Gasser and Grundfest, 1939; Hursh, 1939; Huxley and Stämpeli, 1949; Rushton, 1951). A number of these features are not yet measurable in humans in vivo. However, seminal electrophysiological work has derived a relationship between axon morphology and conduction velocity using only axon diameter and myelin sheath thickness (Rushton, 1951). These two metrics are particularly key because a larger axon diameter results in less resistance to the action potential ion flow, resulting in faster conduction velocity. The presence of a myelin sheath around an axon is beneficial in two ways. First, the myelin sheath acts like an electrical insulating layer, reducing ion loss and preserving the action potential. Second, the presence of unmyelinated gaps in the myelin sheath (the nodes of Ranvier) enables a process called saltatory propagation to take place. As the majority of the axon is wrapped in myelin, the nodes of Ranvier are the only locations where action potentials can occur. This increases the strength of electrical signals because all the ions gather at these nodes instead of being dispersed along the length of the axon. Stronger action potentials are therefore sent along the myelinated portion of the axon at higher speeds, with this signal being boosted on arrival at the next node of Ranvier by another action potential, which helps to maintain a fast conduction velocity.

The conduction velocity of an axon is, therefore, not determined only by the axon diameter, but also by the relationship between the axon diameter and the thickness of the surrounding myelin sheath, a measure known as the g-ratio (Chomiak and Hu, 2009; Rushton, 1951; Schmidt and Knösche, 2019). Specifically, Rushton (1951) derived an equation: conduction velocity∝d−ln(g), where d is the inner axon diameter, g is the g-ratio = d/D, and D is the outer fibre (axon plus myelin sheath) diameter. In other words, the g-ratio is computed as the ratio of the inner axon diameter relative to that of the axon plus the myelin sheath that encases it (Figure 1).”

– On page 7, there is a bit of a leap between the section on the uses and potentials of the MRI-based g-ratio and the proposed application in autobiographical memory. It would help to provide a couple of sentences to link those together or to rearrange a bit the paragraph in-between pages 7 and 8, where at the very end the case for g-ratio and memory is made.

We have now finessed the text further to provide a smoother link.

Introduction, p. 7: “The MR g-ratio has, therefore, the potential to provide a number of novel insights into human cognition. One area where the MR g-ratio may be particularly helpful is in probing individual differences. Our particular interest is in the ability to recall past experiences from real life, known as autobiographical memories.”

– The considerations in figure 1 made on pages 4-5 are at the moment very hard without referring to an actual relationship between conduction velocity, axonal diameter, and g-ratio. Even though the authors are not actually computing conduction velocity, it would help to refer to established formulas such as the one from Rushton (Rushton, Journal of Physiology 1951) or the one form Waxman and Bennet (Waxman and Bennet, Nature New Biology 1972).

We now refer explicitly to the Rushton (1951) formulae for calculating conduction velocity.

Introduction, p. 4: “The conduction velocity of an axon is, therefore, not determined only by the axon diameter, but also by the relationship between the axon diameter and the thickness of the surrounding myelin sheath, a measure known as the g-ratio (Chomiak and Hu, 2009; Rushton, 1951; Schmidt and Knösche, 2019). Specifically, Rushton (1951) derived an equation: conduction velocity∝d−ln(g), where d is the inner axon diameter, g is the g-ratio = d/D, and D is the outer fibre (axon plus myelin sheath) diameter. In other words, the g-ratio is computed as the ratio of the inner axon diameter relative to that of the axon plus the myelin sheath that encases it (Figure 1).”

– I noticed that in Figures 4 and 5 the scatterplots are based on the residuals rather than the actual g-ratio and neurite dispersion values. Can I ask what is the reason behind this choice? This representation makes actually harder to get a visual sense of the dynamics of the reported relationships.

The scatterplots are based on the residuals because we included a number of covariates – age, gender, scanner, number of voxels in the ROI – in the correlation analyses. Plotting the actual g-ratio (or neurite dispersion) values against autobiographical memory recall would not present an accurate plot of the relationship, as calculated with the conservative statistics we employed.

– On page 18, the authors mentioned how these results can provide a new perspective on the underlying neural substrate of autographical memory, and stated: "This is especially welcome given the dearth of consistent findings linking hippocampal grey matter volume or microstructure with autobiographical memory recall ability in the healthy population". Can the authors expand their argument here? I am missing the link between the current findings (on white matter microstructure) with previous ones on grey matter features.

This sentence was simply meant to highlight the fact that we currently lack a clear biological explanation for individual differences in autobiographical memory recall in the healthy population. We have now re-worded this to better reflect the point being made.

p. 19: “These results offer a new perspective on individual differences in autobiographical memory recall ability. This is especially welcome given the current lack of a clear biological explanation for such variations in the healthy population (Clark et al., 2020; Clark et al., 2021a; LePort et al., 2012; Maguire et al., 2003; Palombo et al., 2018; Van Petten, 2004).”

– On page 22, there is a reference to the CONNECTOM MRI scanners (misspelled as "connectome MRI scanners") – as the general audience is not familiar with those it would help to add a brief sentence explaining what they are and maybe a reference (for example, Jones et al., NeuroImage 2018).

We have now included an explanation of Connectom MRI scanners.

Introduction, pp. 5-6: “While measures of myelination can be obtained using conventional MRI scanners (e.g. via magnetisation transfer saturation maps), estimates of inner axon diameter cannot, and instead require ultra-strong gradient systems – 300 mT/m compared to the 10s of mT/m of conventional MRI scanners (Jones et al., 2018; Veraart et al., 2020). However, only four ultra-strong gradient Connectom MRI scanners exist in the world. Consequently, measuring the MR g-ratio with conventional MRI scanners can be useful in guiding inferences about inner axon diameter.”

– Can the authors comment on the advantages of the proposed ROI-based approach compared with a tract-based approach?

We now mention the advantages of the ROI compared to a tract-based approach in the Materials and methods.

Materials and methods, p. 34: “We used the well-established ROI approach (e.g. Ellerbrock and Mohammadi, 2018; Memel et al., 2020) rather than, for example, a tract-based pipeline, in order to reduce the influence of seed and target region selection. This is because small changes to these selections can result in different tracts being identified. We also wanted to avoid the inclusion of excess grey matter in the tracts themselves, because it is not possible to estimate the MR g-ratio in grey matter tissue. In addition, an ROI approach reduced the potential for false positives in comparison to performing tract-based voxel-wise analyses.”

– I may have missed it, but I am not sure of what is the reason for the fornix tract being "not probabilistic" (page 29) – are not all the tract ROIs obtained from the John Hopkins probabilistic atlas?

Thank you for noting this. The fornix was defined using the ICBM-DTI-81 white-matter labels atlas (Mori et al., 2008, NeuroImage), which is a non-probabilistic atlas. The uncinate fasciculus and parahippocampal cingulum bundle were defined using the Johns Hopkins probabilistic atlas. This latter atlas does not include the fornix. The manuscript text has been updated accordingly.

Materials and methods, p. 33: “Bilateral tract ROIs were defined using the Johns Hopkins probabilistic white matter tractography atlas (Hua et al., 2008), with the exception of the fornix which was defined using the ICBM-DTI-81 white-matter labels atlas (Mori et al., 2008) as the fornix is not available in the probabilistic atlas.”

Figure 3 legend: “The three white matter tracts of interest, given their relationship with the hippocampal region. The fornix was defined using the ICBM-DTI-81 white-matter labels atlas (Mori et al., 2008). The uncinate fasciculus and parahippocampal cingulum bundle were defined using the Johns Hopkins probabilistic white matter tractography atlas (Hua et al., 2008), with the minimum probability threshold set to 25%”.

– I could not find any mention of the main methodological limitations (e.g., the use of signal fractions rather than volume fractions, the bias towards larger axons, and the need for single-point calibration). There is only a brief mention in the appendix on page 43. I think these limitations should be part of the discussion to put these results in the right context.

The main methodological limitations are now outlined in the revised manuscript.

Discussion, p. 25: “Nevertheless, as with all neuroimaging techniques, methodological limitations need to be considered when measuring the MR g-ratio (see Campbell et al., 2018; Mohammadi and Callaghan, 2021 for in depth methodological reviews). First, the MR g-ratio is an area-weighted average of all microscopic axons in an MRI-voxel that is slightly weighted towards larger axons. Second, MR proxies are required to estimate myelin and axonal volumes for the calculation of the MR g-ratio. Multiple methodologies are available to estimate myelin and axonal volumes, with no consensus yet reached as to the best combination (Mohammadi and Callaghan, 2021). Third, a calibration step is required to more closely align the MR proxies with the estimated volume fractions. Here, we used the standard single-point calibration method to estimate the slope of the myelin-based proxy, assuming that the offset can be neglected. Future work may be able to improve these calibrations. But, of note, since a non-negligible offset in the myelin-based proxy can increase the error in the MR g-ratio, any calibration improvements would likely serve to increase the observed correlation. Fourth, although the model of the MR g-ratio allows for fibre dispersion (Stikov et al., 2015), this is not accounted for in mapping to conduction velocity. Going forward, more advanced models may help to further elaborate on these relationships. Finally, our myelin measure can be influenced by any factor, cognitive or demographic, that leads to a difference in absolute myelination. In principle, unexplained variance could obscure a true underlying relationship with autobiographical memory. To mitigate this, our analyses controlled for age, gender and any potential scanner-related differences. The MR g-ratio, however, is less affected by this limitation because it is a relative measure and does not depend on absolute myelin content but on the balance between absolute myelin and axonal volumes.”

Reviewer #3 (Recommendations for the authors):I have some suggestions the authors may consider, presented in chronological order.TitleThe title suggests a causal relationship, while this is based on an association. Maybe better to use a neutral word, i.e. replace "influences" with e.g. "is associated with" or something similar?As I understand MR g-ratio, it reflects structural aspects of nerve fibers, and although probably related to conduction velocity/ signaling speed, it is not a direct measure of this. Hence, I believe the title is overstating this relationship by referring to "signalling speed" directly. I fully understand the need for short and focused titles, but I believe this is a bit misleading. The same point applies to a couple of other places in the manuscript too. Of course fine to discuss this as an interpretation.

The title has now been revised to replace “signalling speed” with Reviewer #2’s suggestion of “conduction velocity”, and Reviewer #3’s suggestion to replace “influences” with “associated with”.

Title, p. 1: “Conduction velocity along a key white matter tract is associated with autobiographical memory recall ability”

Any references to causal relationships have also been removed throughout the manuscript.

AbstractNot all readers are probably familiar with the MR g-ratio, so maybe it can be defined in the abstract?

A definition of the g-ratio has now been included.

Abstract, p. 2: “Inferences about conduction velocity can now be made in vivo in humans using a measure called the magnetic resonance (MR) g-ratio. This is the ratio of the inner axon diameter relative to that of the axon plus the myelin sheath that encases it.”

Since this is a cross-sectional study, I suggest avoiding phrases that imply change, e.g. "increased MR g-ratio". Maybe replace it with something neutral, such as "higher MR g-ratio"? The same is the case in several places in the manuscript, e.g. ln 195.As with the title, the results reflect associations, not causation. This does not render them less interesting, but I would recommend replacing "favour better memory …" with e.g. "is associated with better memory …". I also wonder whether "offer a new perspective on drivers of individual differences in memory recall ability" is overstating a bit, since (a) we don't know whether g-ratio (or signal speed) is driving anything, and (b) this is a rather well-established theory (as also explained in the introduction, and it is stated ln 44 "it is perhaps no surprise that speed may be of the essence")?

Throughout the manuscript we have replaced phrases that imply change with more neutral associative statements, and generally tempered the language.

Introductionln 51-54: Why was aging discussed? Reading the introductory paragraph, I expected aging to be part of the study, which it was not.

The literature examining conduction velocity in relation to individual differences is relatively sparse, especially as most work in this area is performed in non-human animals, and involves ageing, which is why we cited it. We have now removed the aging-related summary sentence at the end of that paragraph to prevent confusion.

Ln 123-125: "While some healthy individuals can recollect decades-old autobiographical memories with great richness and clarity, others struggle to recall what they did last weekend" Yes, but as the authors surely know, this reflects different memory systems or processes, and therefore can happen within the same individual too, not only as a feature of inter-individual differences.

We do not deny that variability in recall occurs within the same person as well across people. However, the current study was set up to examine individual differences across people, and the two references cited in relation to that statement (LePort et al., 2012; Palombo et al., 2015) also refer to inter-individual differences.

Ln 195-200: Figure 1 clearly illustrates different scenarios, which is very useful. However, the hypothesis that better autobiographical memory is related to more myelin could also be consistent with no relationship with the MR g-ratio if the axon diameter is larger in participants with a good memory, right? And vice versa for the hypothesis about axon diameter? I don't see how the ratio measure can distinguish between the different accounts since the measures per definition are related to two possibly independent properties which both can be of importance? The authors multiple times mention that other relevant metrics, such as those derived from DTI, lack biological specificity. But the MR g-ratio measure seems also to be unspecific in this regard.

As we now note in the revised manuscript, the conduction velocity of an axon is not determined by the axon diameter or myelin sheath thickness individually, but by the relationship between them – the g-ratio. Given that the MR g-ratio encompasses information about the inner axon diameter and myelin thickness, the identification of statistically significant relationships with the MR g-ratio can help to guide as to which of these two features might be associated variations in conduction velocity. One way to further increase interpretability is by examining scans optimised to assess myelination (such as magnetisation transfer saturation values), as suggested by Reviewer #2. We observed no relationship between autobiographical memory recall ability and magnetisation transfer saturation values. This finding is supportive of the parahippocampal cingulum bundle inner axon diameter being of potentially more relevance than myelin thickness. Consequently, by measuring the MR g-ratio and magnetisation transfer saturation, we can make biologically-relevant inferences about the relationship between autobiographical memory recall ability and conduction velocity. This is in contrast to metrics derived from DTI. In the revised manuscript, we now explain that fractional anisotropy, for example, is a very general measure. Variations in fractional anisotropy values can occur for numerous reasons, including, but not limited to, changes in the extent of myelination, axon coherence, axon density, the level of astrocytes, and so forth.

Introduction, p. 6: “Adjudicating between the possible influence of inner axon diameter or myelination relies on knowledge, or an assumption, about whether the associated change in conduction velocity is faster or slower. This is because significant associations with the MR g-ratio only indicate the existence of a relationship with conduction velocity but not the direction. As noted previously, faster conduction velocity is often held to promote better cognition (e.g. Brancucci, 2012; Dicke and Roth, 2016; Miller, 1994; Reed and Jensen, 1992). Therefore, in Figure 2A, inner axon diameter, myelin thickness and (MR) g-ratio are plotted together to illustrate how different changes in the MR g-ratio are related to the underlying microstructural properties, given a faster conduction velocity. Myelin thickness is represented by the gradient in background colour and contours, with thinnest myelin at the bottom right, and thickest on the top left. The direction of the arrows describes the change in g-ratio for the microstructural variations presented. There are three main scenarios. (1) A decrease in MR g-ratio values. This would suggest that faster conduction velocity is due to greater thickness of the myelin sheath, with the inner axon diameter remaining constant (Figure 2A blue arrow, and Figure 2B). (2) A decrease in MR g-ratio values, but to a lesser extent than that observed in the first scenario. This would suggest that faster conduction velocity is due primarily to greater myelin sheath thickness, but one that is also accompanied by a larger inner axon diameter (Figure 2A red arrow, and Figure 2C). (3) An increase in MR g-ratio values. This would suggest that faster conduction velocity is predominantly due to a larger inner axon diameter, with only small differences, if any, in myelin thickness being present (Figure 2A orange arrow, and Figure 2D).”

Introduction, pp. 9-10: “For completeness, the commonly reported physical parameters from standard diffusion tensor imaging (DTI; e.g. fractional anisotropy and mean diffusivity; Basser, 1995) that are often derived from diffusion data were also computed (Oeschger et al., 2021). However, these metrics lack biological specificity (Jensen and Helpern, 2010; Jones et al., 2013a) and, consequently, could not speak to our research questions. For example, fractional anisotropy is a very general measure. Variations in fractional anisotropy values can occur for numerous reasons, including, but not limited to, changes in the extent of myelination, axon coherence, axon density, and the level of astrocytes.”

Discussion, pp. 19-20: “A key property of the MR g-ratio is that it provides a non-invasive methodology with which to relate in vivo structural neuroimaging of humans with axonal conduction velocity (Berman et al., 2019; Drakesmith et al., 2019). Moreover, the identification of a statistically significant relationship with the MR g-ratio can help to guide as to which of inner axon diameter or myelin thickness is more likely to be associated with variations in conduction velocity (Caeyenberghs et al., 2016; Kaller et al., 2017; Lakhani et al., 2016; Waxman, 1980; Xin and Chan, 2020).

Our inferences rested upon the assumption that a significant relationship with the MR g-ratio reflected faster conduction velocity, given previous work suggesting that faster conduction velocity might be associated with better cognition (e.g. Brancucci, 2012; Dicke and Roth, 2016; Miller, 1994; Reed and Jensen, 1992). As shown in Figure 2, there are three main scenarios describing how changes in the MR g-ratio are related to the underlying microstructural properties, given a faster conduction velocity. The positive relationship we observed between the parahippocampal cingulum bundle MR g-ratio and autobiographical memory recall ability suggests that this effect was associated predominantly with larger inner axon diameter (Figure 2A orange arrow, and Figure 2D). By contrast, had a negative correlation between the MR g-ratio and autobiographical memory recall ability been identified, we could instead have inferred that myelination was the relevant microstructural feature. The lack of relationship between parahippocampal cingulum bundle magnetisation transfer saturation values (optimised to assess myelination) and autobiographical memory recall ability provided further corroboration that myelin may not have been influential in this context. Greater myelination is often held to be a prominent influence on behavioural and cognitive performance (Caeyenberghs et al., 2016; Fields and Bukalo, 2020; Kaller et al., 2017; Lakhani et al., 2016; Waxman, 1980; Xin and Chan, 2020). By contrast, our results highlight the potentially important role that the inner axon diameter could be playing in autobiographical memory recall.”

If axon diameter and myelin thickness can be estimated separately, why not use them directly in the analyses instead of the MR g-ratio? There may be a methodological reason for this, but I think this could be more clearly explained as MR g-ratio is not often reported in the literature and so will be novel to many readers.

As we now note in the revised manuscript, the conduction velocity of an axon is not determined by the axon diameter or myelin sheath thickness individually, but by the relationship between them – the g-ratio. The MR g-ratio, therefore, is the only measure able to provide a link with conduction velocity.

While measures of myelination can be obtained using conventional MRI scanners (e.g. via the magnetisation transfer saturation values that we also acquired), estimates of inner axon diameter cannot, and instead require ultra-strong gradient systems – 300 mT/m in comparison to the 10s of mT/m of conventional MRI scanners (Jones et al., 2018; Veraart et al., 2020). However, only four ultra-strong gradient Connectom MRI scanners exist in the world. Consequently, measuring the MR g-ratio and interpreting these results together with magnetisation transfer saturation values can be useful in guiding inferences about inner axon diameter with conventional MRI scanners.

Introduction, p. 4: “The conduction velocity of an axon is, therefore, not determined only by the axon diameter, but also by the relationship between the axon diameter and the thickness of the surrounding myelin sheath, a measure known as the g-ratio (Chomiak and Hu, 2009; Rushton, 1951; Schmidt and Knösche, 2019). Specifically, Rushton (1951) derived an equation: conduction velocity∝d−ln(g), where d is the inner axon diameter, g is the g-ratio = d/D, and D is the outer fibre (axon plus myelin sheath) diameter. In other words, the g-ratio is computed as the ratio of the inner axon diameter relative to that of the axon plus the myelin sheath that encases it (Figure 1).”

Introduction, p. 5: “Of note, the MR g-ratio has been found to associate well with estimates of axonal conduction velocity (Berman et al., 2019; Drakesmith et al., 2019). Consequently, the MR g-ratio provides a non-invasive MRI method that can associate in vivo structural neuroimaging of humans with axonal conduction velocity.”

Introduction, p. 5-6: “Given that the MR g-ratio encompasses information about the inner axon diameter and myelin thickness, the identification of statistically significant relationships with the MR g-ratio can also guide as to which of these two features might be influencing variations in conduction velocity (Caeyenberghs et al., 2016; Kaller et al., 2017; Lakhani et al., 2016; Waxman, 1980; Xin and Chan, 2020). This is particularly useful in relation to the inner axon diameter. While measures of myelination can be obtained using conventional MRI scanners (e.g. via magnetisation transfer saturation maps), estimates of inner axon diameter cannot, and instead require ultra-strong gradient systems – 300 mT/m compared to the 10s of mT/m of conventional MRI scanners (Jones et al., 2018; Veraart et al., 2020). However, only four ultra-strong gradient Connectom MRI scanners exist in the world. Consequently, measuring the MR g-ratio with conventional MRI scanners can be useful in guiding inferences about inner axon diameter.”

Discussion, pp. 19-20: “A key property of the MR g-ratio is that it provides a non-invasive methodology with which to relate in vivo structural neuroimaging of humans with axonal conduction velocity (Berman et al., 2019; Drakesmith et al., 2019). Moreover, the identification of a statistically significant relationship with the MR g-ratio can help to guide as to which of inner axon diameter or myelin thickness is more likely to be associated with variations in conduction velocity (Caeyenberghs et al., 2016; Kaller et al., 2017; Lakhani et al., 2016; Waxman, 1980; Xin and Chan, 2020).”

Relatedly, although there are methods papers on this topic, maybe a brief justification of how can ratios between these small compartments on the micrometer scale can be calculated from voxels on the millimeter scale already in the introduction?

We now briefly expand this explanation in the revised manuscript.

Introduction, p. 5: “Until recently, g-ratio measurements were restricted to invasive studies in non-human animals. However, by combining diffusion magnetic resonance imaging (MRI) with quantitative structural MRI scans optimised to assess myelination (e.g. magnetisation transfer saturation; Weiskopf et al., 2013), it is now possible to estimate the g-ratio in vivo in humans across the whole brain (Drakesmith et al., 2019; Mohammadi et al., 2015; Mohammadi and Callaghan, 2021; Stikov et al., 2015). This is achieved by measuring an aggregate g-ratio, which is an area-weighted ensemble average across a voxel of an underlying distribution of microscopic g-ratios of axons (Stikov et al., 2015; West et al., 2016).”

ResultsI understand that the authors don't want to provide the full description of results for FA and MD in the main text, but it would be convenient to report a brief summary of the main results also in the text. As far as I can see from Appendix 1, none even approach significance for fornix and uncinated fasciculus, so maybe at least state that?

Brief summaries of the standard DTI parameters are now provided in the Results section, and are mentioned in the Discussion.

Results, pp. 13-14: “Standard DTI parameters (e.g. fractional anisotropy and mean diffusivity) were also extracted from the fornix and uncinate fasciculus. None of the standard DTI parameters, from either tract, were significantly associated with autobiographical memory recall ability, even when using an uncorrected p < 0.05 threshold (see Appendix 1–tables 1-4 for full details, source data are available in Supplementary file 1).”

Results, pp. 17-18:

“Standard DTI parameters of the parahippocampal cingulum bundle and autobiographical memory recall ability

Partial correlations using the same covariates as the previous analyses were also performed using the standard DTI parameters. As these metrics lack biological specificity (Jensen and Helpern, 2010; Jones et al., 2013a), and could not speak to our research questions, the results are summarised here, with full details in Appendix 1–tables 19 and 20 and Appendix 1–figures 9 and 10, and the source data are available in Supplementary file 2.

Two significant relationships were identified between the standard DTI parameters and autobiographical memory recall ability. First, there was a positive correlation between fractional anisotropy and the number of internal details (Appendix 1–figure 9A; r(211) = 0.20, p = 0.003, 95% CI = 0.07, 0.32). Second, a positive correlation was evident between diffusivities parallel and the number of internal details (Appendix 1–figure 10A; r(211) = 0.19, p = 0.005, 95% CI = 0.06, 0.32). As with the microstructure metrics, these relationships were specific to internal details with no relationships identified with external details for either fractional anisotropy (r(211) = -0.06, p = 0.39, 95% CI = -0.19, 0.07) or diffusivities parallel (r(211) = -0.048, p = 0.49, 95% CI = -0.19, 0.10). No significant relationships were found between mean diffusivity, mean kurtosis or diffusivities perpendicular and either internal or external details.”

Discussion, p. 22: “There have only been a small number of previous studies investigating white matter tracts and individual differences in autobiographical memory recall ability in healthy people. These focused on standard DTI parameters, such as fractional anisotropy and mean diffusivity, and not the MR g-ratio. Two such studies were suggestive of a relationship between parahippocampal cingulum bundle fractional anisotropy and autobiographical memory recall (Irish et al., 2014; Memel et al., 2020). Our analyses of the standard DTI parameters, therefore, replicated these previous results.”

Discussion, p. 23: “In terms of standard DTI parameters, unlike a previous study (Hodgetts et al., 2017), we did not find a relationship between fornix fractional anisotropy and autobiographical memory recall ability, which may be due to the larger sample that we examined.”

This may be a matter of taste, but I would suggest moving the section on why the parahippocampal cingulum bundle should be important for autobiographical memory from the Results section (ln 236 and onwards) to the introduction or the discussion.

We understand the Reviewer’s point, and did consider this idea before the original submission. We have elected to keep this section within the Results to act as a primer for readers when they reach the parahippocampal cingulum bundle findings, to remind them about why this tract may be especially relevant. However, consideration of the parahippocampal cingulum bundle and its possible role in autobiographical memory recall is also included in both the Introduction and Discussion.

Ln 279: A positive relationship between MR g-ratio and memory is interpreted as reflecting larger inner axon diameters. Ref Figure 1, this can also be due to thinner myelin sheets, right, as it is a ratio measure?

A positive relationship between MR g-ratio and autobiographical memory recall would only reflect thinner myelin sheaths if the conduction velocity was slower (rather than faster, which is the basis of our interpretations). In addition, we have now also included analyses with a measure of myelination (magnetisation transfer saturation) and found no relationship between myelin and autobiographical memory recall ability. This supports our suggestion that the MR g-ratio finding might be related to inner axon diameter rather than myelination, and consequently also the assumption of faster rather than slower conduction velocity.

Multiple comparison correction was mentioned for some analyses above, was this done here two, and if so, how? Would be useful to state explicitly.

Multiple comparison correction was performed for all partial correlation analyses using the Bonferroni method; dividing α = 0.05 by 3 (because there were three tracts of interest). This means that associations with a two-sided p-value < 0.017 were considered significant. This information is included in the Materials and methods on p. 35.

Inspecting the Appendix, it seems that the relationships between autobiographic memory and PCB FA and MD were also significant, with r-values that were at least as large as for MR g-ratio. Why not include this in the main text? These are metrics of interest to many researchers.

Brief summaries of the standard DTI parameters are now provided in the Results section, and are mentioned in the Discussion.

Results, pp. 17-18:

“Standard DTI parameters of the parahippocampal cingulum bundle and autobiographical memory recall ability

Partial correlations using the same covariates as the previous analyses were also performed using the standard DTI parameters. As these metrics lack biological specificity (Jensen and Helpern, 2010; Jones et al., 2013a), and could not speak to our research questions, the results are summarised here, with full details in Appendix 1–tables 19 and 20 and Appendix 1–figures 9 and 10, and the source data are available in Supplementary file 2.

Two significant relationships were identified between the standard DTI parameters and autobiographical memory recall ability. First, there was a positive correlation between fractional anisotropy and the number of internal details (Appendix 1–figure 9A; r(211) = 0.20, p = 0.003, 95% CI = 0.07, 0.32). Second, a positive correlation was evident between diffusivities parallel and the number of internal details (Appendix 1–figure 10A; r(211) = 0.19, p = 0.005, 95% CI = 0.06, 0.32). As with the microstructure metrics, these relationships were specific to internal details with no relationships identified with external details for either fractional anisotropy (r(211) = -0.06, p = 0.39, 95% CI = -0.19, 0.07) or diffusivities parallel (r(211) = -0.048, p = 0.49, 95% CI = -0.19, 0.10). No significant relationships were found between mean diffusivity, mean kurtosis or diffusivities perpendicular and either internal or external details.”

Discussion, p. 22: “There have only been a small number of previous studies investigating white matter tracts and individual differences in autobiographical memory recall ability in healthy people. These focused on standard DTI parameters, such as fractional anisotropy and mean diffusivity, and not the MR g-ratio. Two such studies were suggestive of a relationship between parahippocampal cingulum bundle fractional anisotropy and autobiographical memory recall (Irish et al., 2014; Memel et al., 2020). Our analyses of the standard DTI parameters, therefore, replicated these previous results.”

You could also run a multiple regression analysis, and test whether each metric gives unique information or whether they are redundant. I think this would be a very useful addition to the manuscript, regardless of the outcome. If MR g-ratio is still significantly related to autobiographical memory, this would strengthen the main argument in the paper. If not, we will know more about the interrelationships between these different tract metrics, which also will be very useful.

Unfortunately, it is not possible to implement this suggestion. Each of the metrics available has varying specificity in relation to the underlying microstructure. Fractional anisotropy, for example, is a very general measure that simply describes the directionality of water molecule movement. High fractional anisotropy values can occur, therefore, for a variety of reasons including increased myelination, axon coherence and axon density. This means that fractional anisotropy will always be able to explain more of the variance than the more specific measures. However, this does not mean it has greater explanatory power, because fractional anisotropy cannot inform about the actual variations in microstructure, while the more specific metrics can.

To illustrate, in the current study, we see relationships between autobiographical memory recall and fractional anisotropy, axonal coherence and the MR g-ratio, and no relationship with axonal density. As the fractional anisotropy metric is the most general measure, placing the four measures together into a regression model results in fractional anisotropy explaining the majority of the model variance. However, even if we had observed instead, for example, relationships between autobiographical memory recall and fractional anisotropy, axonal *density* and the MR g-ratio, and no relationship with axonal coherence, exactly the same result would emerge from the regression analysis. Fractional anisotropy would again seem to explain the majority of the model variance, even though different inferences would be drawn from investigating the individual measures alone.

DiscussionLn 336: Is it correct to say that the MR g-ratio – memory correlation represents "neural instantiation of autobiographical memory"? To me, it seems to overstate the results, as this is an association, and we do not know whether there is any causality involved. The same goes for the last part of the sentence "drivers of individual differences in recall ability". I think the results are of sufficient interest, without such statements.

These statements have now been removed.

Methods, ParticipantsLn 456-457: "People with hobbies or vocations known to be associated with the hippocampus (e.g. licenced London taxi drivers) were excluded." Could you please be a little more specific, most activities will be associated with the hippocampus? How many were excluded based on this?

Additional information has now been included in the revised manuscript.

Materials and methods, pp. 26-27: “Our aim was to assess people from the general population who would not be classed as having extreme expertise on classic hippocampal tasks, as this could affect hippocampal structure (Maguire et al., 2000; Woollett and Maguire, 2011). Consequently, people with vocations such as taxi driving (or those training to be taxi drivers), ship navigators, aeroplane pilots, or those with regular hobbies including orienteering, or taking part in memory sports and competitions, were excluded. Of the approximately 2000 people who contacted us, 23 were explicitly excluded on this basis.”

[Editors’ note: what follows is the authors’ response to the second round of review.]

The authors put an amazing effort into taking into account the reviewers' comments and into improving the paper with further details and analysis. The finding that magnetization transfer saturation does not correlate with the score of interest while the g-ratio instead does is quite interesting -

Thank you.

I think that for the sake of fully characterizing the relationship what is missing at the moment is correlating the isotropic volume fraction with the recall score: since the g-ratio is computed using MTsat, Viso and Vic, double-checking that a correlation does not exist with Viso is the last step needed to show that indeed is the combination of all these measures that is meaningful rather than the single measures themselves.

Author response: Thank you for this suggestion. The correlation between autobiographical memory recall ability and Viso (isotropic volume fraction) has now been computed and included in the manuscript (and the associated data are now included in the relevant supplementary file). As expected, no significant relationship was observed, confirming that it is the combination of the measures, in the form of the MR g-ratio, that is meaningful rather than any single measure alone. We have modified the text as follows:

Results, pp. 17-18: “For completeness, we also examined a third measure from the NODDI biophysical model, the isotropic volume fraction. This models the space occupied by cerebrospinal fluid in a voxel, and therefore no relationship with autobiographical memory recall ability was expected, and none was found. Partial correlations showed that the isotropic volume fraction of the parahippocampal cingulum bundle (mean = 0.04, SD = 0.03) was not significantly related to either internal (r(211) = -0.03, p = 0.71, 95% CI = -0.14, 0.10), or external (r(211) = 0.02, p = 0.77, 95% CI = -0.11, 0.15) details. This confirms that it is the combination of these measures, in the form of the MR g-ratio, that is meaningful, rather than any single measure alone.”

Discussion, p. 25: “Relationships with autobiographical memory recall ability were also specific to the MR g-ratio, and were not evident for any of its components (magnetisation transfer saturation, neurite density or isotropic volume fraction), suggesting that it is the combination of these measures, in the form of the MR g-ratio, that is meaningful rather than any single measure alone.”